# On Affine Homotopy between Language Encoders

**Robin S. M. Chan**[1] **Reda Boumasmoud**[1] **Anej Svete**[1] **Yuxin Ren**[2]
**Qipeng Guo**[3] **Zhijing Jin**[1,4] **Shauli Ravfogel**[1] **Mrinmaya Sachan**[1]
**Bernhard Schölkopf**[1,4] **Mennatallah El-Assady**[1] **Ryan Cotterell**[1]
[1]ETH Zürich  [2]Tsinghua University  [3]Fudan University
[4]Max Plank Institute for Intelligent Systems

## Abstract

Pre-trained language encoders—functions that represent text as vectors—are an integral component of many NLP tasks. We tackle a natural question in language encoder analysis: What does it mean for two encoders to be similar? We contend that a faithful measure of similarity needs to be *intrinsic*, that is, task-independent, yet still be informative of *extrinsic* similarity—the performance on downstream tasks. It is common to consider two encoders similar if they are *homotopic*, i.e., if they can be aligned through some transformation.[1] In this spirit, we study the properties of *affine* alignment of language encoders and its implications on extrinsic similarity. We find that while affine alignment is fundamentally an asymmetric notion of similarity, it is still informative of extrinsic similarity. We confirm this on datasets of natural language representations. Beyond providing useful bounds on extrinsic similarity, affine intrinsic similarity also allows us to begin uncovering the structure of the space of pre-trained encoders by defining an order over them.

○ https://github.com/chanr0/affine-homotopy

## 1 Introduction

A common paradigm in modern natural language processing (NLP) is to pre-train a **language encoder** on a large swathe of natural language text. Then, a task-specific model is fit (*fine-tuned*) using the language encoder as the representation function of the text. More formally, a language encoder is a function $\boldsymbol{h} \colon \Sigma^* \to \mathbb{R}^D$, i.e., a function that maps a string over an alphabet $\Sigma$ to a finite-dimensional vector. Now, consider sentiment analysis as an informative example of a task. Suppose our goal is to classify a string $\boldsymbol{y} \in \Sigma^*$ as one of three polarities $\Pi = \{\odot, \odot, \odot\}$. Then, the probability of $\boldsymbol{y}$ exhibiting a specific polarity is often given by a log-linear model, e.g., the probability of $\odot$ is

$$p(\odot \mid \boldsymbol{y}) = \mathrm{softmax}(\mathbf{E}\,\boldsymbol{h}(\boldsymbol{y}) + \mathbf{b})_{\odot} \tag{1}$$

where $\mathbf{E} \in \mathbb{R}^{3 \times D}$, $\mathbf{b} \in \mathbb{R}^3$ and $\mathrm{softmax} \colon \mathbb{R}^N \to \Delta^{N-1}$. Empirically, using a pre-trained encoder $\boldsymbol{h}$ leads to significantly better classifier performance than training a log-linear model from scratch.

In the context of the widespread deployment of language encoders, this paper tackles a natural question: Given two language encoders $\boldsymbol{h}$ and $\boldsymbol{g}$, how can we judge to what extent they are similar? This question is of practical importance—recent studies have shown that even small variations in the random seed used for training can result in significant performance differences on downstream tasks between models with the same architecture [13, 35] In this case, we say that two such language encoders exhibit an *extrinsic* difference, i.e., the difference between two encoders manifests itself when considering their performance on a *downstream* task. However, we also seek an *intrinsic* notion

---

[1]Homotopy, from the Greek ὁμός (homo; same) and τόπος (topos; place), refers to a continuous transformation between functions or shapes, showing they can be deformed into one another without breaking or tearing.

38th Conference on Neural Information Processing Systems (NeurIPS 2024).

of similarity between two language encoders, i.e., a notion of similarity that is independent of any particular downstream task. Moreover, we may hope that a good notion of intrinsic similarity would allow us to construct a notion of extrinsic similarity that holds for *all* downstream tasks.

Existing work studies language encoder similarity by evaluating whether two encoders produce similar representations for a finite dataset of strings [3, 20, 22, 42, *inter alia*], often by analyzing whether the representation sets can be approximately *linearly* aligned [22, 27]. More formally, two encoders are considered similar if there exists a matrix $\mathbf{A}$ such that $\boldsymbol{h}(\boldsymbol{y}) \approx \mathbf{A}\,\boldsymbol{g}(\boldsymbol{y})$ holds for strings $\boldsymbol{y}$ in some finite set $\mathcal{D} \subset \Sigma^*$.[2] This assumes that examining finitely many outputs provides sufficient insight into encoder behavior. In contrast, we set out to study the relationships between language encoders, i.e., functions, themselves. This decision, rather than being just a technicality, allows us to derive a richer understanding of encoder relationships, revealing properties and insights that remain obscured under conventional finite-set analysis. Concretely, we ask what notions of similarity between encoders one could consider and what they imply for their relationships.

The main contributions of the paper are of a theoretical nature. We first define an (extended) metric space on language encoders. We then extend this notion to account for *transformations* in a broad framework of $S$-**homotopy** for a set of transformations $S$, where $\boldsymbol{g}$ is $S$-homotopic to $\boldsymbol{h}$ if $\boldsymbol{g}$ can be transformed into $\boldsymbol{h}$ through some transformation in $S$. As a concrete application of the framework, we study *affine* homotopy—the similarity of $\boldsymbol{h}$ and $\psi \circ \boldsymbol{g}$ for affine transformations $\psi$. The notion of intrinsic similarity induced by such one-sided alignment is not symmetric and can be seen as the *cost* of transforming $\boldsymbol{g}$ into $\boldsymbol{h}$. Nevertheless, we show it is informative of *extrinsic* similarity: If one encoder can be affinely mapped to another, we can guarantee that it also performs similarly on downstream tasks. We confirm this empirically by studying the intrinsic and extrinsic similarities of various pretrained encoders, where we observe a positive correlation between intrinsic and extrinsic similarity. Beyond measuring similarity, homotopy also allows us to define a form of hierarchy on the space of encoders, elucidating a structure in which some encoders are more informative than others. Such an order is also suggested by our experiments, where we find that certain encoders are easier to map to than others which shows in the rank of the learned representations and affects their transfer learning ability.

## 2    Language Encoders

Let $\Sigma$ be an alphabet—a finite, non-empty set of symbols $y$—and $\text{EOS} \notin \Sigma$ a distinguished end-of-string symbol. With $\Sigma^* \overset{\text{def}}{=} \bigcup_{n=0}^{\infty} \Sigma^n$ we denote the Kleene closure of $\Sigma$, the set of all strings $\boldsymbol{y}$. A **language encoder** is a function $\boldsymbol{h} \colon \Sigma^* \to V \overset{\text{def}}{=} \mathbb{R}^D$ that maps strings to real vectors.[3] We write $\mathcal{E}_V \overset{\text{def}}{=} V^{\Sigma^*}$ for the $\mathbb{R}$-vector space of language encoders, and $\mathcal{E}_b \overset{\text{def}}{=} \{\boldsymbol{h} \in \mathcal{E}_V \mid \boldsymbol{h}(\Sigma^*) \text{ is bounded}\} \subset \mathcal{E}_V$ for its sub-vector space of **bounded encoders**.

There are two common ways that language encoders are created [7]. The first is through autoregressive language modeling. A **language model** (LM) is a probability distribution over $\Sigma^*$.[4] **Autoregressive** LMs are defined through the multiplication of conditional probability distributions $p_{\boldsymbol{h}}(y_t \mid \boldsymbol{y}_{<t})$ as

$$p_{\boldsymbol{h}}^{\text{LM}}(\boldsymbol{y}) = p_{\boldsymbol{h}}(\text{EOS} \mid \boldsymbol{y}) \prod_{t=1}^{T} p_{\boldsymbol{h}}(y_t \mid \boldsymbol{y}_{<t}), \tag{2}$$

where each $p_{\boldsymbol{h}}(\cdot \mid \boldsymbol{y}_{<t})$ is a distribution over $\Sigma \cup \{\text{EOS}\}$ *parametrized* by a language encoder $\boldsymbol{h}$:

$$p_{\boldsymbol{h}}(y_t \mid \boldsymbol{y}_{<t}) \overset{\text{def}}{=} \text{softmax}(\mathbf{E}\,\boldsymbol{h}(\boldsymbol{y}_{<t}))_{y_t}, \tag{3}$$

where $\mathbf{E} \in \mathbb{R}^{(|\Sigma|+1) \times D}$. An autoregressive LM provides a simple manner to *learn* a language encoder from a dataset of strings $\mathcal{D} = \{\boldsymbol{y}^{(n)}\}_{n=1}^{N}$ by minimizing $\mathcal{D}$'s negative log-likelihood. We may also learn a language encoder through **masked language modeling** (MLM), which defines the conditional probabilities based on both sides of the masked symbol's context

$$p_{\boldsymbol{h}}(y_t \mid \boldsymbol{y}_{<t}, \boldsymbol{y}_{>t}) \overset{\text{def}}{=} \text{softmax}(\mathbf{E}\,\boldsymbol{h}(\boldsymbol{y}_{<t} \circ [\text{MASK}] \circ \boldsymbol{y}_{>t}))_{y_t}. \tag{4}$$

---

[2]We discuss related work in more detail in App. C.

[3]In principle, one could relax the replace $\mathbb{R}^D$ with any finite dimensional vector space.

[4]In the following, we assume language model *tightness* to the effect that we can assume that LMs produce valid probability distributions over $\Sigma^*$ [15].

Maximizing the log-likelihood of a corpus under a language model derived from a language encoder $h$ with a gradient-based algorithm only requires $h$ to be a differentiable function of its parameters. Once a language encoder has been trained on a (large) corpus, its representations can be used on more fine-grained NLP tasks such as classification. The rationale for such transfer learning is that representations $h(y)$ stemming from a performant language model also contain information useful for other downstream tasks on natural language. An NLP practitioner might then implement a task-specific transformation of $h(y)$. To tackle the problem that the tasks of interest are often less resource-abundant and to keep the training costs low, task-specific transformations are usually simple, often in the form of linear transformations of $h(y)$, as in Eq. (1).

## 3 Measuring the Alignment of Langauge Encoders

We begin by introducing measures of affine alignment and hemi-metrics on $\mathcal{E}_V$.

### 3.1 Preliminaries on Hemi-Metric Spaces

Language encoders compute representations for the infinitely many strings in $\Sigma^*$. In general, these representations might diverge towards $\infty$, making it necessary to talk about *unbounded* encoders, where it is convenient to allow distances and norms to take extended real numbers as values.[5]

**Definition 3.1.** *An **extended metric** on a set $X$ is a map $d\colon X \to \overline{\mathbb{R}}_+$ such that*

    *a.* $\forall x, y \in X, \quad d(x, y) = 0$ *iff* $x = y$;            *(Identity)*

    *b.* $\forall x, y, z \in X, \quad d(x, y) \leqslant d(x, z) + d(z, y)$;        *(Triangle Inequality)*

    *c.* $\forall x, y \in X, \quad d(x, y) = d(y, x)$.            *(Symmetry)*

Similarly, an **extended norm** is a map $\|\cdot\|\colon X \to \overline{\mathbb{R}}_+$ that satisfies the norm axioms. Moreover, we will consider maps $d$ that do not satisfy the symmetry axiom. Lawvere [25] notes that symmetry is artificial and unnecessary for many of the main theorems involving metric spaces. In such situations, the quantity $d(x, y)$ can be interpreted as the *cost* of going from $x$ to $y$. Occasionally, we want $d$ to capture that it costs more to go from $x$ to $y$ than to return, making asymmetry desirable.

**Definition 3.2.** *A **hemi-metric**[6] or Lawvere-metric on a set $X$ is a map $d\colon X \to \overline{\mathbb{R}}_+$ such that*

    *a.* $d(x, x) = 0$,

    *b.* $d(x, z) \leqslant d(x, y) + d(y, z)$       *for all $x, y, z \in X$.*

One of our main contributions is a formalization of measuring how far a language encoder $h$ is from the *set* of all possible transformations of another encoder $g$—for example, from all affine transformations of $g$. For this, we *lift* a hemi-metric over elements $x \in X$ to *subsets* of $X$, a crucial for the rest of the paper.

**Definition 3.3.** *Let $(X, d)$ be a hemi-metric space. For non-empty $E, E' \subset X$, we define*

$$d^{\mathcal{H}}(E, E') \stackrel{\text{def}}{=} \sup_{x \in E} \inf_{y \in E'} d(x, y). \tag{5}$$

*The map $d^{\mathcal{H}}$ is called the **Hausdorff–Hoare map** and is a hemi-metric on $\mathcal{P}(X) \backslash \{\varnothing\}$, the power set of $X$. When $E$ is a singleton set $\{x\}$, we will, with a slight abuse of notation, write $d^{\mathcal{H}}(x, E')$ to mean $d^{\mathcal{H}}(\{x\}, E')$, defined as $= \inf_{y \in E'} d(x, y)$.[7]*

We next introduce the **hemi-metric recipe**. It tells us how one can define a hemi-metric on a set $X$ by *embedding* $X$ into the power set of another space $Y$ where a hemi-metric already exists. After $X$ is embedded, one can use the Hausdorff–Hoare map based on the hemi-metric from $Y$ to define a hemi-metric on $X$ through the images of $x \in X$.

---

[5]$\overline{\mathbb{R}}_+$ is the set of non-negative real numbers along with the "value" $\infty$, assumed to be above all reals. We adopt the following conventions: $\infty \cdot 0 = 0 \cdot \infty = 0$; $\infty + r = r + \infty = \infty$, $\infty \cdot r = \mathbb{R}_{>0}$ for every $r \in \mathbb{R}_{>0}$.

[6]A basic example of a hemi-metric space is the pair $(\mathbb{R}, d_{\mathbb{R}})$, where $d_{\mathbb{R}}(x, y) = \max(x - y, 0)$.

[7]Additional properties of $d^{\mathcal{H}}$ are discussed in Lem. D.1.

**Remark 3.1** (Hemi-Metric Recipe). *Let $X$ be a set, $(Y, d)$ a hemi-metric space, and $S\colon X \to \mathcal{P}(Y)\backslash\{\varnothing\}, x \mapsto E_x$ a function that assigns an $x \in X$ a subset $E_x \in \mathcal{P}(Y)\backslash\{\varnothing\}$. Using Lem. D.1, we can construct a hemi-metric on $X$ with $d_S^{\mathcal{H}}(x,y) \overset{\text{def}}{=} d^{\mathcal{H}}(E_x, E_y) = d^{\mathcal{H}}(S(x), S(y))$, and an* **extended pseudo-metric** *(a symmetric hemi-metric) with $d_S^{\mathcal{H},\, sym}(x,y) = \max(d_S^{\mathcal{H}}(x,y), d_S^{\mathcal{H}}(y,x))$.*

Remark 3.1 introduces a general recipe for defining hemi-metric spaces on function spaces—topological spaces whose elements are functions from a set to subsets of an extended-metric space. This naturally applies to the study of encoders and their transformations, which we call $S$-**homotopy**, i.e., two encoders are $S$-**homotopic** if one can be $S$-deformed into the other. In this case, the set $E_{\boldsymbol{h}}$ for $\boldsymbol{h} \in \mathcal{E}_V$ corresponds to the set of encoders that $\boldsymbol{h}$ can be transformed into with mappings in $S$. We could, for example, take $S$ as the set of all continuous maps, smooth maps, or multi-layer perceptrons. Our following discussion of affine maps, i.e., where $S = \mathrm{Aff}(V)$, is extrinsically motivated but can be understood as a specific instance of the more general framework of $S$-homotopy.

### 3.2 A Norm and a Distance on $\mathcal{E}_V$

The hemi-metric recipe first requires us to define a (hemi-)metric on the individual elements. Given that all norms on the $\mathbb{R}$-vector space $V$ are equivalent [24, Proposition 2.2, §XII], we fix in this paper a norm $|\cdot|_V$ on $V$. We introduce the maps $\|\cdot\|_\infty\colon \mathcal{E}_V \to \overline{\mathbb{R}}_+$ and $d_\infty(\cdot,\cdot)\colon \mathcal{E}_V \times \mathcal{E}_V \to \overline{\mathbb{R}}_+$:

$$\|\boldsymbol{h}\|_\infty \overset{\text{def}}{=} \sup_{\boldsymbol{y} \in \Sigma^*} |\boldsymbol{h}(\boldsymbol{y})|_V \qquad (6) \qquad \text{and} \qquad d_\infty(\boldsymbol{h}, \boldsymbol{g}) = \|\boldsymbol{h} - \boldsymbol{g}\|_\infty, \qquad (7)$$

where $\|\cdot\|_\infty$ is an extended norm on $\mathcal{E}_V$ and $(\mathcal{E}_V, d_\infty)$ is a complete[8] extended metric space.[9]

Let $\mathrm{GL}(V)$ be the set of invertible $D \times D$ matrices. We write $\|\cdot\|_V\colon \mathrm{GL}(V) \to \mathbb{R}_+$ for the subordinate matrix norm, i.e., $\|\mathbf{A}\|_V = \sup_{\mathbf{v} \in V\backslash\{0\}} \frac{|\mathbf{Av}|_V}{|\mathbf{v}|_V}$. By abuse of language, we can view $V$ as an affine space[10] and set $\mathrm{Aff}(V)$ for the group of affine transformations of $V$. An affine transformation $\psi$ on $V$ is a map $\mathbf{v} \mapsto \mathbf{Av} + \mathbf{b}$, for some invertible $\mathbf{A} \in \mathrm{GL}(V)$ and $\mathbf{b} \in V$. We call $\psi_{\text{lin}} \overset{\text{def}}{=} \mathbf{A}$ the linear part of $\psi$ and $t_\psi\colon \mathbf{v} \mapsto \mathbf{v} + \mathbf{b}$ its translation part. We denote with $\mathcal{T} \subset \mathrm{Aff}(V)$ the subgroup of translations. Note that there is a natural left action of $\mathrm{Aff}(V)$ on $\mathcal{E}_V$, i.e., $\mathrm{Aff}(V) \times \mathcal{E}_V \to \mathcal{E}_V, \boldsymbol{h} \mapsto \psi \circ \boldsymbol{h}$.[11]

### 3.3 Affine Alignment of Language Encoders

We now use the general recipe from Remark 3.1 for *affine alignment* of language encoders—affinely mapping from one encoder to another. For a subset $S \subset \mathrm{Aff}(V)$ we can define

$$d_S(\boldsymbol{h}, \boldsymbol{g}) \overset{\text{def}}{=} d_\infty^{\mathcal{H}}(\boldsymbol{h}, S(\boldsymbol{g})) = \inf_{\psi \in S} \|\boldsymbol{h} - \psi \circ \boldsymbol{g}\|_\infty \qquad (8a)$$

$$\|\boldsymbol{h}\|_S \overset{\text{def}}{=} d_S(0_{\mathcal{E}_V}, \boldsymbol{h}), \qquad (8b)$$

where $S(\boldsymbol{h}) \overset{\text{def}}{=} \{s \circ \boldsymbol{h} \mid s \in S\}$. In the notation of the hemi-metric recipe from Remark 3.1, we set $X = Y = \mathcal{E}_V$ (we align an encoder with another encoder) and $d = d_\infty$, the uniform convergence distance (cf. §3.2). Further, we take $S \subseteq \mathrm{Aff}(V) \subset V^V$ and define $S\colon \mathcal{E}_V \to \mathcal{P}(\mathcal{E}_V), \boldsymbol{h} \mapsto S(\boldsymbol{h}) \overset{\text{def}}{=} \{s \circ \boldsymbol{h} \mid s \in S\}$. In words, $d_S(\boldsymbol{h}, \boldsymbol{g})$ captures the notion of how well the encoder $\boldsymbol{g}$ can be $S$-transformed into $\boldsymbol{h}$. This is commonly called the **alignment** of $\boldsymbol{g}$ with $\boldsymbol{h}$. $d_S(\boldsymbol{h}, \boldsymbol{g})$ does not, however, necessarily tell us anything about how well $\boldsymbol{h}$ can be $S$-transformed into $\boldsymbol{g}$, resulting in asymmetry.

**Remark 3.2.** $d_{\mathrm{Aff}(V)}$ *defined in Eq.* (8a) *is not a metric on $\mathcal{E}_V$.*[12] *Further, when $S = \mathrm{Aff}(V)$, the map $\inf_{\psi, \psi' \in \mathrm{Aff}(V)} \|\psi \circ \boldsymbol{h} - \psi' \circ \boldsymbol{g}\|_\infty$ is trivially zero by Cor. D.1.*

---

[8]A metric space is complete if every Cauchy sequence (a sequence where the distance between terms eventually becomes arbitrarily small) converges to a point within the space.

[9]This follows immediately from the fact that $|\cdot|_V$ is a norm and from the completeness of $V$.

[10]This amounts to "forgetting" the special role played by the zero vector.

[11]A left action of a group $G$ on a set $X$ is a map $\cdot\colon G \times X \to X$ such that $e \cdot x = x$ for all $x \in X$, where $e$ is the identity element of $G$, and $g_1 \cdot (g_2 \cdot x) = (g_1 g_2) \cdot x$ for all $g_1, g_2 \in G$ and $x \in X$.

[12]See App. D.2 for a derivation.

In the case of affine isometries $\mathrm{Iso}(V) = \{\psi \in \mathrm{Aff}(V) \mid \psi_{\mathrm{lin}} \in \mathrm{O}(V)\}$ we show that the pair $(\mathcal{E}_V, d_{\mathrm{Iso}(V)})$ constitutes an extended pseudo-metric space.

**Proposition 3.1.** *The pair $(\mathcal{E}_V, d_{\mathrm{Iso}(V)})$ is an extended pseudo-metric space.*

# 4 Intrinsic Affine Homotopy

The notion of affine alignment allows us to introduce *homotopic relations* on $\mathcal{E}_V$. We first derive the affine intrinsic preorder $\succsim_{\mathrm{Aff}}$ on the space of encoders.[13]

**Lemma 4.1.** *Let $(X, d)$ be a hemi-metric space. The relation $(x \succsim_d y$ iff $d(x,y) = 0)$ is a **preorder**[14] and it will be called the **specialization ordering** of $d$.*

*Proof.* Goubault-Larrecq [17, Proposition 6.1.8]. ∎

**Definition 4.1** (Intrinsic Affine Preorder). *For two encoders $h, g \in \mathcal{E}_V$, we define the relation*

$$h \succsim_{\mathrm{Aff}} g \quad \text{iff} \quad d_{\mathrm{Aff}(V)}(h, g) = 0. \tag{9}$$

**Lemma 4.2.** *The relation $\succsim_{\mathrm{Aff}}$ is a preorder on $\mathcal{E}_V$.*

*Proof.* Follows from $d_{\mathrm{Aff}(V)}(\psi \circ h, g) \leqslant \|\psi_{\mathrm{lin}}\|_V \cdot d_{\mathrm{Aff}(V)}(h, g)$, see App. D.3. ∎

Intuitively, $\succsim_{\mathrm{Aff}}$ captures the order of encoders such that higher-positioned encoders in the order can be $S$-transformed to the lower-positioned ones. To derive the implications of $\succsim_{\mathrm{Aff}}$ we introduce the notion of an encoder rank.

**Definition 4.2** (Encoder Rank). *For any $h \in \mathcal{E}_V$ let the **encoder rank** be $\mathrm{rank}(h) \overset{\text{def}}{=} \dim_{\mathbb{R}}(V_h)$, where $V_h$ is the subvector space generated by the image of $h$. When $\mathrm{rank}(h) = \dim_{\mathbb{R}}(V)$, $h$ is a **full rank** encoder, else it is **rank deficient**.*

**Theorem 4.1.** *For $h, g \in \mathcal{E}_V$, we have*

$$h \succsim_{\mathrm{Aff}} g \Leftrightarrow h = \psi(\pi_h \circ g) \text{ for some } \psi \in \mathrm{Aff}(V) \tag{10}$$

*where, $\pi_h$ is the orthogonal projection of $V$ onto $V_h$. In particular, if $d_{\mathrm{Aff}(V)}(h, g) = 0$ then $\mathrm{rank}(h) \leqslant \mathrm{rank}(g)$. If in addtion, we know $\mathrm{rank}(g) = \mathrm{rank}(h)$, then $g$ must by an affine transformation of $g$, i.e., $h = \psi \circ g$ for some $\psi \in \mathrm{Aff}(V)$.*

This allows us to state our first notion of language encoder similarity: intrinsic affine homotopy.

**Definition 4.3** (Exact Intrinsic Affine Homotopy). *We say that two encoders $h, g \in \mathcal{E}_V$ are **exactly intrinsically affinely homotopic** and write $h \simeq_{\mathrm{Aff}} g$ if*

$$d_{\mathrm{Aff}(V)}(h, g) = 0 \text{ and } \mathrm{rank}(h) = \mathrm{rank}(g). \tag{11}$$

For any $h, g \in \mathcal{E}_V$, one can easily show that

$$h \simeq_{\mathrm{Aff}} g \iff (g \succsim_{\mathrm{Aff}} h \text{ and } h \succsim_{\mathrm{Aff}} g) \iff d_{\mathrm{Aff}(V)}^{\mathcal{H}, \mathrm{sym}}(h, g) = 0, \tag{12}$$

which implies that $\simeq_{\mathrm{Aff}}$ is an equivalence relation on the set of language encoders $\mathcal{E}_V$. Intuitively, two encoders $h$ and $g$ are exactly intrinsically affinely homotopic, this means that both $g$ can be affinely mapped to $h$, as well as the other way around.

# 5 Extrinsic Homotopy

In §4, we explore methods for assessing how similar two language encoders are without reference to any downstream tasks. Here, we extend our discussion to the *extrinsic* homotopy of language encoders. Since language encoders are primarily used to generate representations for downstream tasks—such as in transfer learning, illustrated by the sentiment analysis example in §1—we argue that the key criterion in the similarity of two encoders lies in how closely we can align predictions stemming from their representations.[15]

---

[13]All left-out proofs can be found in App. D.2.

[14]A reflexive and transitive relation on $X$.

[15]The proofs of all claims in this section can be found in App. D.3.

**Principle 5.1** (Extrinsic Homotopy). Two language encoders $\boldsymbol{h}$ and $\boldsymbol{g}$ are **extrinsically homotopic** if we can guarantee a similar performance on any downstream task $\boldsymbol{h}$ and $\boldsymbol{g}$ might be used for.

The rest of the section formalizes this intuitive notion and describes its relationship with intrinsic affine homotopy. Let $W$ be the vector space $\mathbb{R}^N$ and set $\mathrm{Aff}(V, W)$ as the set of affine maps from $V$ to $W$.[16] We define $\mathcal{E}_\Delta \overset{\mathrm{def}}{=} \mathrm{Map}(\Sigma^*, \Delta^{N-1})$ and $\mathcal{E}_W = \mathrm{Map}(\Sigma^*, W)$. Lastly, we formalize the notion of a transfer learning task as constructing a classifier that uses a language encoder's string representations. Particularly, we set $\mathcal{V}_N$ to be the family of log-linear models as follows

$$\mathcal{V}_N \colon \mathcal{E}_V \to \mathcal{P}(\mathcal{E}_{\Delta^{N-1}}) \backslash \{\varnothing\}, \quad \boldsymbol{h} \mapsto \mathrm{softmax}_\lambda(\mathrm{Aff}_{V,W}(\boldsymbol{h})), \tag{13}$$

where $\mathrm{Aff}_{V,W}$ is the map

$$\mathrm{Aff}_{V,W} \colon \mathcal{E}_V \to \mathcal{P}(\mathcal{E}_W) \backslash \{\varnothing\}, \quad \boldsymbol{h} \mapsto \{\psi \circ \boldsymbol{h} \mid \psi \in \mathrm{Aff}(V, W)\} \tag{14}$$

and $\mathrm{softmax}_\lambda \colon \mathbb{R}^N \to \Delta^{N-1}$ is defined for $\lambda \in \mathbb{R}_+$, $\mathbf{x} \in \mathbb{R}^N$, and $n \in [N]$ as

$$\mathrm{softmax}_\lambda(\mathbf{x})_n = \frac{\exp(\lambda x_n)}{\sum_{n'=1}^N \exp(\lambda x_{n'})}. \tag{15}$$

**Remark 5.1.** *Each $p_\psi = \mathrm{softmax}_\lambda \circ \psi(\boldsymbol{h}(\boldsymbol{y}))$ can be seen as a "probability distribution" over $[N]$*

$$\mathcal{V}_N(\boldsymbol{h}) = \{p(\square \mid \boldsymbol{y}) \colon [N] \to [0, 1], \square \mapsto \mathrm{softmax}_\lambda \circ \psi(\boldsymbol{h}(\boldsymbol{y}))_\square \mid \psi \in \mathrm{Aff}(V, W)\}. \tag{16}$$

Through our standard recipe from Remark 3.1, we can define the following hemi-metrics on $\mathcal{E}_V$.

**Definition 5.1.** *For any two encoders $\boldsymbol{h}, \boldsymbol{g} \in \mathcal{E}_V$, we define*[17]

$$d_{\mathrm{Aff}(V,W)}^{\mathcal{H}}(\boldsymbol{h}, \boldsymbol{g}) \overset{\mathrm{def}}{=} d_{\infty, W}^{\mathcal{H}}(\mathrm{Aff}_{V,W}(\boldsymbol{h}), \mathrm{Aff}_{V,W}(\boldsymbol{g})) \tag{17a}$$

$$d_{\mathcal{V}(V,\Delta)}^{\mathcal{H}}(\boldsymbol{h}, \boldsymbol{g}) \overset{\mathrm{def}}{=} d_{\infty, \Delta^{N-1}}^{\mathcal{H}}(\mathcal{V}_N(\boldsymbol{h}), \mathcal{V}_N(\boldsymbol{g})) \tag{17b}$$

Notice that we use $d^{\mathcal{H}}$ rather than $d$ in Def. 5.1 since we are interested in how closely we can bring $\boldsymbol{h}$ and $\boldsymbol{g}$ when we affinely transform *both* of them—this corresponds to independently affinely transforming the encoders for the same transfer learning task. In particular, Eq. (17b) measures how different two encoders are on any transfer learning task, formalizing the notion of extrinsic homotopy (cf. Principle 5.1), captured by the following definition.

**Definition 5.2** (Extrinsic Affine Preorder). *An encoder $\boldsymbol{h} \in \mathcal{E}_V$ is **exactly extrinsically homotopic** to*[18] *$\boldsymbol{g} \in \mathcal{E}_V$ if $d_{\mathcal{V}(V,\Delta)}^{\mathcal{H}}(\boldsymbol{h}, \boldsymbol{g}) = 0$.*

Analogously to Def. 4.1, we use $d_{\mathcal{V}(V,\Delta)}^{\mathcal{H}}(\boldsymbol{h}, \boldsymbol{g})$ to define a preorder.

**Definition 5.3** (Extrinsic Affine Preorder). *For two encoders $\boldsymbol{h}, \boldsymbol{g} \in \mathcal{E}_V$, we define the relation*

$$\boldsymbol{h} \succsim_{\mathrm{Ext}} \boldsymbol{g} \quad \mathit{iff} \quad d_{\mathcal{V}(V,\Delta)}^{\mathcal{H}}(\boldsymbol{h}, \boldsymbol{g}) = 0. \tag{18}$$

**Lemma 5.1.** *The relation $\boldsymbol{h} \succsim_{\mathrm{Ext}} \boldsymbol{g}$ is a preorder on $\mathcal{E}_V$.*

We now relate $d_{\mathrm{Aff}(V,W)}^{\mathcal{H}}(\boldsymbol{h}, \boldsymbol{g})$ and $d_{\mathcal{V}(V,\Delta)}^{\mathcal{H}}(\boldsymbol{h}, \boldsymbol{g})$ from Def. 5.1, and $d_{\mathrm{Aff}(V)}(\boldsymbol{h}, \boldsymbol{g})$ from §4.

**Lemma 5.2.** *Let $\boldsymbol{h}, \boldsymbol{g} \in \mathcal{E}_V$. We have*

1. *There exists a constant $c(\lambda) > 0$ such that for any $\psi \in \mathrm{Aff}(V, W)$*

$$d_{\infty, \Delta^{N-1}}^{\mathcal{H}}(\mathrm{softmax}_\lambda(\psi \circ \boldsymbol{h}), \mathcal{V}_N(\boldsymbol{g})) \leqslant c(\lambda) \|\psi_{lin}\| d_{\mathrm{Aff}(V)}(\boldsymbol{h}, \boldsymbol{g}).$$

2. *$d_{\mathcal{V}(V,\Delta)}^{\mathcal{H}}(\boldsymbol{h}, \boldsymbol{g}) \leqslant c(\lambda) d_{\mathrm{Aff}(V,W)}^{\mathcal{H}}(\boldsymbol{h}, \boldsymbol{g})$.*

---

[16]Given an affine map $f \colon V \to W$, there is a unique linear map $\mathbf{A} = f_{\mathrm{lin}} \in \mathcal{L}(V, W)$ and $\mathbf{b} \in W$ such that for every $v \in V$ we have $f(\mathbf{v}) = \mathbf{A} \cdot \mathbf{v} + \mathbf{b}$.

[17]The subscript $\infty$ in $d_{\infty, \Delta}$ and $d_{\infty, W}$ is used to insist on that we are considering the supremum distance $d_\infty$ in $\Delta^{N-1}$ and $W$, respectively.

[18]Exact extrinsic homotopy is asymmetric.

3. $d_{\mathrm{Aff}(V)}(\boldsymbol{h},\boldsymbol{g}) = 0 \Rightarrow d^{\mathcal{H}}_{\mathrm{Aff}(V,W)}(\boldsymbol{h},\boldsymbol{g}) = 0 \Rightarrow d^{\mathcal{H}}_{\mathcal{V}(V,\Delta)}(\boldsymbol{h},\boldsymbol{g}) = 0.$

Lem. 5.2 shows that $\succsim_{\mathrm{Ext}}$ is *finer* than $\succsim_{\mathrm{Aff}}$. This means that the affine intrinsic preorder is contained in the extrinsic preorder, i.e., $\boldsymbol{h} \succsim_{\mathrm{Aff}} \boldsymbol{g} \Rightarrow \boldsymbol{h} \succsim_{\mathrm{Ext}} \boldsymbol{g}$. Lastly, we can show that $d^{\mathcal{H}}_{\mathcal{V}(V,\Delta)}(\boldsymbol{h},\boldsymbol{g})$ is upper bounded by the intrinsic hemi-metric $d^{\mathcal{H}}_{\mathrm{Aff}(V)}$.

**Theorem 5.1** ($\epsilon$-Intrinsic $\Rightarrow \mathcal{O}(\epsilon)$-Extrinsic)**.** *Let $\boldsymbol{h},\boldsymbol{g} \in \mathcal{E}_V$ be two encoders. Then,*

$$d^{\mathcal{H}}_{\mathcal{V}(V,\Delta)}(\boldsymbol{h},\boldsymbol{g}) \leqslant c(\lambda)\, d^{\mathcal{H}}_{\mathrm{Aff}(V)}(\boldsymbol{h},\boldsymbol{g}).$$

# 6 Linear Alignment Methods for Finite Representation Sets

§§ 4 and 5 introduce ways of comparing language encoders as functions, which holistically characterizes relationships between them. We now address a more practical concern: Given two language encoders $\boldsymbol{h}$ and $\boldsymbol{g}$, how can we approximate their similarity in practice? Rather than comparing $\boldsymbol{h}(\boldsymbol{y}) : \Sigma^* \to \mathbb{R}^D$ with $\boldsymbol{g}(\boldsymbol{y}) : \Sigma^* \to \mathbb{R}^D$ over the entire $\Sigma^*$,[19] we compare them over a finite set of strings $\mathcal{Y} = \{\boldsymbol{y}^{(n)}\}_{n=1}^N$. We combine $\mathcal{Y}$'s representations given by $\boldsymbol{h}$ and $\boldsymbol{g}$ into matrices $\mathbf{H}, \mathbf{G} \in \mathbb{R}^{N \times D}$, where we denote $\mathbf{H}_{\boldsymbol{y},\cdot} = \boldsymbol{h}(\boldsymbol{y})$ and $\mathbf{G}_{\boldsymbol{y},\cdot} = \boldsymbol{g}(\boldsymbol{y})$. We can approximate the notions of similarity from §3 by optimizing over the affine maps $\mathrm{Aff}(V)$ (for example, using gradient descent). Particularly, we approximate intrinsic similarity as

$$\hat{d}_{\mathrm{Aff}(V)}(\mathbf{H},\mathbf{G}) \stackrel{\text{def}}{=} \inf_{\psi \in \mathrm{Aff}(V)} \max_{\boldsymbol{y} \in \mathcal{Y}} \|\mathbf{H}_{\boldsymbol{y},\cdot} - \psi \circ \mathbf{G}_{\boldsymbol{y},\cdot}\|_V, \tag{19}$$

and extrinsic similarity for some task-specific fixed $\psi'$ as

$$\hat{d}_{\psi'}(\mathbf{H},\mathbf{G}) \stackrel{\text{def}}{=} \inf_{\psi \in \mathrm{Aff}(V,W)} \max_{\boldsymbol{y} \in \mathcal{Y}} \|\mathrm{softmax}(\psi' \circ \mathbf{H}_{\boldsymbol{y},\cdot}) - \mathrm{softmax}(\psi \circ \mathbf{G}_{\boldsymbol{y},\cdot})\|_W. \tag{20}$$

Unfortunately, the $\max$ over $\mathcal{Y}$ makes the optimization in Eqs. (19) and (20) difficult. For simplicity, we turn to commonly used linear alignment methods, which we review for completeness.

**Orthogonal Procrustes Problem.** Rather than optimizing the infinity norm over $\mathcal{Y}$ as Eqs. (19) and (20), the orthogonal Procrustes problem finds the orthogonal transformation minimizing the Frobenius norm [34] by solving $\operatorname{argmin}_{\mathbf{A} \in \mathrm{O}(V)} \|\mathbf{H} - \mathbf{A}\mathbf{G}\|_F$. Given the singular-value decomposition $\mathbf{H}^\top \mathbf{G} = \mathbf{U}\boldsymbol{\Sigma}\mathbf{V}^\top$, the optimum is achieved by $\mathbf{U}\mathbf{V}^\top$.[20] Since the $\operatorname{argmin}$ is over $\mathrm{O}(V)$, this defines an extended pseudo-metric space by Prop. 3.1.

**Canonical Correlation Analysis (CCA).** CCA [20] is a linear alignment method that finds the matrices $\mathbf{A}, \mathbf{B}$ that project $\mathbf{H}$ and $\mathbf{G}$ into subspaces maximizing their canonical correlation. Let $\mathbf{A}_{\cdot,j}$ and $\mathbf{B}_{\cdot,j}$ be the $j$th column vectors of $\mathbf{A}$ and $\mathbf{B}$, respectively. The formulation is as follows

$$\rho_j = \sup_{\mathbf{A}_{\cdot,j},\mathbf{B}_{\cdot,j}} \mathrm{corr}(\mathbf{H}\mathbf{A}_{\cdot,j}, \mathbf{G}\mathbf{B}_{\cdot,j}) \quad \text{s.t.} \quad \forall_{i<j}\ \mathbf{H}\mathbf{A}_{\cdot,j} \perp \mathbf{H}\mathbf{A}_{\cdot,i}, \quad \forall_{i<j}\ \mathbf{G}\mathbf{B}_{\cdot,j} \perp \mathbf{G}\mathbf{B}_{\cdot,i}. \tag{21}$$

The representation similarity is measured in terms of the goodness of CCA fit, e.g., the mean squared CCA correlation $R^2_{\mathrm{CCA}} = \sum_{i=1}^D \rho_i^2 / D$. We can reformulate the CCA objective in Eq. (21) as

$$\inf_{\mathbf{A},\mathbf{B}} \frac{1}{2} \|\mathbf{A}^\top \mathbf{H} - \mathbf{B}^\top \mathbf{G}\|_F^2 \quad \text{s.t.} \quad (\mathbf{A}^\top \mathbf{H})(\mathbf{A}^\top \mathbf{H})^\top = (\mathbf{B}^\top \mathbf{G})(\mathbf{B}^\top \mathbf{G})^\top = \mathbf{I}. \tag{22}$$

Given the singular-value decomposition $\mathbf{H}^\top \mathbf{G} = \mathbf{U}\boldsymbol{\Sigma}\mathbf{V}^\top$, the solution of Eq. (22) is $(\hat{\mathbf{A}}, \hat{\mathbf{B}}) = ((\mathbf{H}\mathbf{H}^\top)^{-\frac{1}{2}}\mathbf{U}, (\mathbf{G}\mathbf{G}^\top)^{-\frac{1}{2}}\mathbf{V})$, where $(\mathbf{H}\mathbf{H}^\top)^{-\frac{1}{2}}$ and $(\mathbf{G}\mathbf{G}^\top)^{-\frac{1}{2}}$ are whitening transforms of $\mathbf{U}$ and $\mathbf{V}$. Assuming the data is whitened during pre-processing, CCA corresponds to linear alignment under an orthogonality constraint, equivalent to the orthogonal Procrustes problem; see also App. E.

**CCA Extensions.** Projection-weighted CCA (PWCCA) [30] also finds alignment matrices with CCA but applies weighting to correlation values $\rho_i$ to report the goodness of fit. Given the canonical vectors $\hat{\mathbf{A}}$, PWCCA reports $\bar{\rho}_{\mathrm{PW}} = \sum_{i=1}^D \alpha_i \rho_i / \sum_i \alpha_i$, where $\alpha_i = \sum_j |\langle \hat{\mathbf{A}}_{\cdot,i}, \mathbf{H}_{\cdot,j} \rangle|$.[21]

---

[19]For simplicity, we assume that $\boldsymbol{h}$ and $\boldsymbol{g}$ both map to $\mathbb{R}^D$

[20]See App. E for the derivation.

[21]CCA extensions beyond PWCCA are dicussed in App. C.

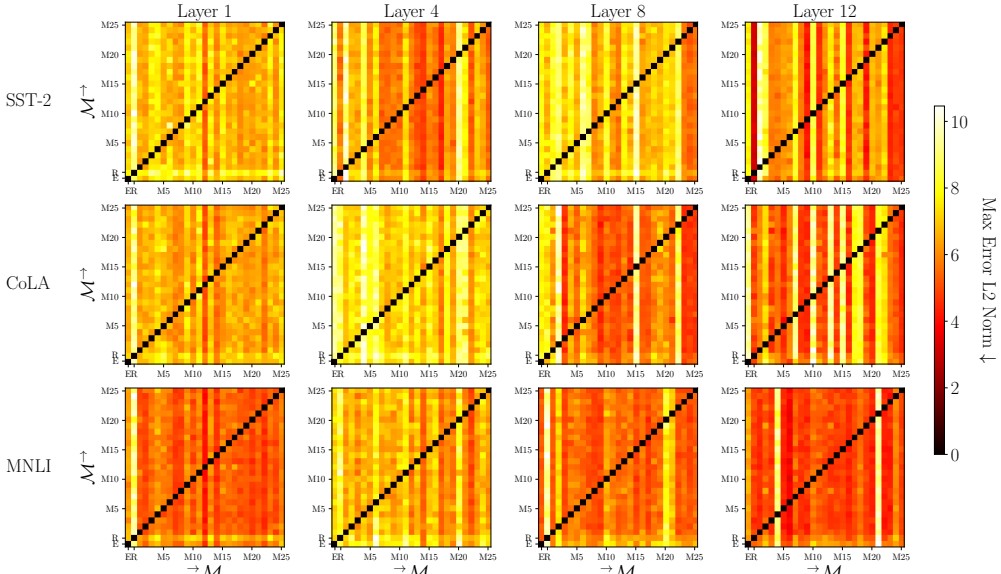

Figure 1: Asymmetry between ELECTRA (E), RoBERTa (R), and MULTIBERT encoders (M1-M25) across layers. For each pair of the encoders $\mathcal{M}^{(i)}$ and $\mathcal{M}^{(j)}$, we generate training set embeddings $\mathbf{H}^{(i)}, \mathbf{H}^{(j)} \in \mathbb{R}^{N \times D}$ for SST-2, COLA, and MNLI. We then fit $\mathbf{H}^{(i)}$ to $\mathbf{H}^{(j)}$ with an affine map and report the goodness of fit through the max error L2 norm, i.e., an approximation of $d(\mathbf{H}^{(j)}, \mathbf{H}^{(i)})$ on row $i$ and column $j$ of the grid. Full results across GLUE tasks are shown in Figure 4.

**Non-Alignment Methods.** While not explicitly (linearly) aligning representations, CKA [22] evaluates the kernel similarity between representations. CKA computes the normalized Hilbert-Schmidt independence [18] between centered kernel matrices $\mathbf{K^H}$ and $\mathbf{K^G}$ where $\mathbf{K}_{ij}^{\mathbf{H}} = k(\mathbf{H}_{i,\cdot}, \mathbf{H}_{j,\cdot})$, and $\mathbf{K}_{ij}^{\mathbf{G}} = k(\mathbf{G}_{i,\cdot}, \mathbf{G}_{j,\cdot})$ for a kernel function $k$, i.e., $\operatorname{tr}(\mathbf{K^H K^G})/\sqrt{(\operatorname{tr}(\mathbf{K^H}^\top \mathbf{K^H})\operatorname{tr}(\mathbf{K^G}^\top \mathbf{K^G}))}$. Linear CKA, where $k(\mathbf{H}_{i,\cdot}, \mathbf{H}_{j,\cdot}) = \mathbf{H}_{i,\cdot}^\top \mathbf{H}_{j,\cdot}$, is commonly used.

## 7 Experiments

We now explore the practical implications of our theoretical results. We conduct experiments on ELECTRA [6], ROBERTA [28], and the 25 MULTIBERT [35] encoders, which are architecturally identical to BERT-BASE [11] models pre-trained with different seeds. We report results on the training sets of two GLUE benchmark classification tasks: SST-2 [38] and MRPC [14]. When reporting $d$ and $\hat{d}_{\psi'}$ from Eq. (19) and Eq. (20), we use the $L_2$ norm for simplicity and approximate $d_{\mathcal{V}(V,\Delta)}^{\mathcal{H}}$ as

$$\hat{d}_{\mathcal{V}(V,\Delta)}^{\mathcal{H}}(\mathbf{H}, \mathbf{G}) = \sup_{\psi' \in \operatorname{Aff}(V,W)} \inf_{\psi \in \operatorname{Aff}(V,W)} \max_{\boldsymbol{y} \in \mathcal{Y}} \|\operatorname{softmax}(\psi' \circ \mathbf{H}_{\boldsymbol{y},\cdot}) - \operatorname{softmax}(\psi \circ \mathbf{G}_{\boldsymbol{y},\cdot})\|_2. \quad (23)$$

The experimental setup and compute resources are further described in App. F.

**The Intrinsic 'Preorder' of Encoders.** We first investigate whether the asymmetry of $d_{\operatorname{Aff}(V)}$ is measurable in the finite alphabet encoder representations. Figure 1 shows distinct vertical lines for both tasks indicating that there are encoders that are consistently easier to affinely map *to* ($^\rightarrow\mathcal{M}$). This seems to be rather independent of which encoder we map *from* ($\mathcal{M}^\rightarrow$). We further see that this trend is task-independent for early layers but diverges for later layers.

**The Influence of Encoder Rank Deficiency.** As discussed in §4, the encoder rank plays a pivotal role in affine mappability; exact affine homotopy is only achievable between equal-rank encoders.[22] With this in mind, we return to our findings from Figure 1 to evaluate whether the observed differences

---

[22]We provide additional experiments on the role of the encoder rank in App. G.

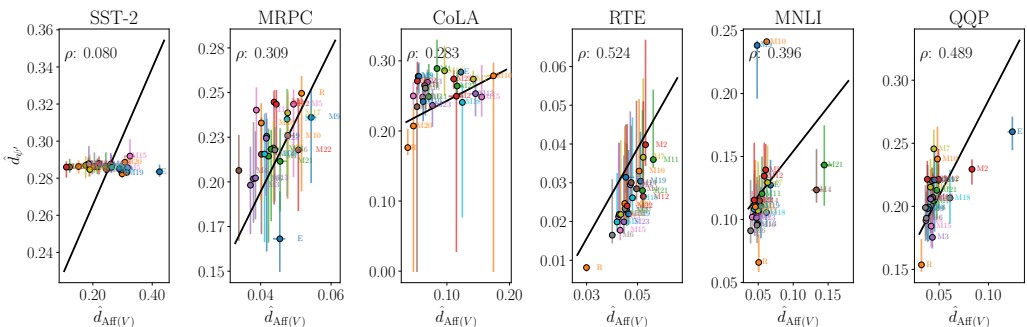

Figure 2: For ELECTRA (E), RoBERTa (R), and MULTIBERTs (M1-M25), we plot extrinsic ($\hat{d}_{\psi'}$) against intrinsic similarity ($\hat{d}_{\text{Aff}(V)}$) across GLUE tasks. We group the points by how well we can map to each encoder ($\rightarrow \mathcal{M}$), and display the median, as well as the first and third quartiles as vertical and horizontal lines. We additionally show the linear regression from $\hat{d}_{\text{Aff}(V)}$ to $\hat{d}_{\psi'}$.

between encoders can be attributed to a difference in measurable rank. Due to the inaccuracies of computing the rank numerically, we approximate the encoder rank using the **rank to precision** $\epsilon$ as the number of representation matrix singular values larger than some $\epsilon \in \mathbb{R}$.[23] We find statistically significant ($p$-value $< 0.05$) rank correlation with the median intrinsic distance $\hat{d}_{\text{Aff}(V)}$ when mapping *to* the corresponding encoder for RTE ($\rho = 0.312$), MRPC ($\rho = 0.609$), and QQP ($\rho = 0.389$). We find no statistically significant correlations with the median distance when mapping *from* the corresponding encoder. This difference in encoder ranks could, therefore, partially explain the previously observed differences in affine mappability as some encoders seem to learn lower-rank representations.

**A Linear Bound on Extrinsic Similarity.** Lem. 5.2 derives a relationship between affine intrinsic and extrinsic similarity. To evaluate its strength in practice, we measure Spearman's Rank Correlation ($\rho$) and Pearson Correlation (PCC) between intrinsic measures introduced in §6 and the extrinsic measures $\hat{d}_{\psi'}$ and $\hat{d}_{\mathcal{V}(V,\Delta)}^{\mathcal{H}}$. PCC measures the strength and direction of a linear relationship between two random variables, whereas Spearman's $\rho$ additionally evaluates the variables' monotonic association. $\hat{d}_{\psi'}$ is computed by training a linear classifier $\psi' \in \text{Aff}(V)$ on the final MULTIBERT layer for each task. Further, we report $\hat{d}_{\mathcal{V}(V,\Delta)}^{\mathcal{H}}$ as the maximum $L_2$ loss for a large number of randomly generated[24] classifiers $\psi'$ on the final layer of each MULTIBERT encoder. We generate 100 such classifiers for a range of GLUE datasets.[25] Table 1 show significant, large linear correlation prevalent in all linear alignment methods, whereas CKA—a linear, non-alignment method—does not capture extrinsic behavior as faithfully. Further, Figure 2 visualizes the linear relationship explicitly for all considered GLUE datasets.

# 8 Discussion

We set out to explore homotopic relationships between language encoders, augmenting existing work on the similarity of finite representation sets by holistically studying encoder *functions*. In particular, the general framework of $S$-homotopy allows us to study any functional relationship between encoders, enabling the exploration of many types of encoder relationships. As a first step in this direction and a concrete example, §4 explores *affine* homotopy, discussing what it means to be able to align two models with affine transformations. Here, Hausdorff–Hoare maps prove useful, as they allow us to measure a notion of (asymmetric) distance between a point—an encoder—and the *set* of all affine transformations of another encoder. Lem. 5.2 in §5 then connects the intrinsic,

---

[23]Following Press et al. [31], we choose $\epsilon$ as $n \cdot \sigma_1 \cdot \epsilon_p \cdot \max(N, D)$ for $n = 5$, where $\sigma_1$ is the largest singular value of the $N \times D$ representation matrix and $\epsilon_p$ the float machine epsilon. We note that the rank to precision $\epsilon$ and the recovered correlation may depend on the chosen $\epsilon$.

[24]The generation process is described in App. F.

[25]The computational expense of computing $\hat{d}_{\mathcal{V}(V,\Delta)}^{\mathcal{H}}$ restricts this analysis to a limited set of classifiers, depending on the alphabet size. See App. B for a discussion.

| Intrinsic Measure | | $\hat{d}_{\text{Aff}(V)}$ | Orth. Procrustes | $R^2_{CCA}$ | PWCCA | Linear CKA |
|---|---|---|---|---|---|---|
| **Lin. Alignment-Based** | | Yes | Yes | Yes | Yes | No |
| SST-2 | $\hat{d}_{\text{Aff}(V)}$ $\rho$ | 0.080 | 0.095 | **0.172*** | 0.016 | 0.088 |
| | PCC | 0.545* | 0.937* | 0.932* | **0.970*** | 0.231* |
| | $\hat{d}^{\mathcal{H}}_{\mathcal{V}(V,\Delta)}$ $\rho$ | **0.621*** | 0.157* | 0.071 | 0.231* | 0.295* |
| | PCC | **0.723*** | 0.539* | 0.457* | 0.566* | 0.320* |
| MRPC | $\hat{d}_{\psi'}$ $\rho$ | **0.309*** | 0.250* | -0.001 | 0.220* | 0.214* |
| | PCC | 0.707* | 0.697* | 0.733* | **0.743*** | 0.241* |
| | $\hat{d}^{\mathcal{H}}_{\mathcal{V}(V,\Delta)}$ $\rho$ | **0.231*** | 0.025* | 0.178* | 0.059 | 0.030 |
| | PCC | 0.790* | 0.755* | **0.879*** | 0.875* | 0.174* |
| RTE | $\hat{d}_{\psi'}$ $\rho$ | **0.534*** | 0.053 | 0.037 | 0.308* | 0.185* |
| | PCC | **0.570*** | 0.401* | 0.429* | 0.250* | 0.078 |
| | $\hat{d}^{\mathcal{H}}_{\mathcal{V}(V,\Delta)}$ $\rho$ | 0.234* | 0.317* | -0.147* | **0.338*** | 0.240* |
| | PCC | 0.718* | **0.870** | 0.778 | 0.780 | 0.205 |
| CoLA | $\hat{d}_{\psi'}$ $\rho$ | **0.196*** | 0.006 | 0.040 | 0.185* | 0.165* |
| | PCC | 0.204* | 0.529* | **0.553*** | 0.550* | 0.215* |
| | $\hat{d}^{\mathcal{H}}_{\mathcal{V}(V,\Delta)}$ $\rho$ | 0.348* | 0.078 | 0.133* | 0.340* | **0.380*** |
| | PCC | 0.429* | 0.664* | 0.318* | **0.786*** | 0.513* |

Table 1: Spearman's Rank Correlation Coefficient ($\rho$) and Pearson's Correlation Coefficient (PCC) between intrinsic measures introduced in §6 and the extrinsic similarities $\hat{d}_{\psi'}$ and $\hat{d}^{\mathcal{H}}_{\mathcal{V}(V,\Delta)}$ across various GLUE datasets. * indicates a $p$-value $< 0.01$ (assuming independence).

task-independent, similarly to extrinsic similarity—the similarity of performance on downstream tasks. Concretely, it derives a linear relationship between the intrinsic and extrinsic dissimilarity for any fixed affine transformation $\psi'$ (i.e., a fixed downstream task). Thm. 5.1 discusses a stronger bound, namely on the *worst-case* extrinsic dissimilarity among all downstream linear classifiers, i.e., among all possible tasks. Further, by accounting for the asymmetries of encoder relationships, we augment the work on similarity in proper metric spaces [3, 36, 42].

Although encoders may not be affinely related in practice, empirical evidence in §7 suggests that notions of affine order still surface (cf. Tab. 1, Fig. 2), particularly as differently initialized BERTs exhibit variations in downstream task performance [29]. While other similarity measures, such as those used in seed specificity tests [12], are designed to remain invariant to initialization changes, our results indicate that intrinsic affine homotopy is appropriately *sensitive* to them. This sensitivity raises new questions about the landscape of pre-trained encoders; as seen in Fig. 1, asymmetry in intrinsic affine similarity among similarly pre-trained encoders impacts downstream performance, as corroborated by Lem. 5.2 and empirical results in Tab. 1. Differences in representation ranks may partly explain this asymmetry—mapping between artificially generated rank-deficient encoders yields mostly symmetric affine distances (cf. Fig. 3). Another explanation might be that easy-to-learn encoders might be approximately linear combinations of others, making them easy to map *to* but not necessarily *from*. Overall, our findings highlight the need to account for directionality in encoder similarity measures to address the asymmetry inherent in this problem.

# 9   Conclusion

We discuss the structure of the space of language encoder in the framework of $S$-homotopy—the notion of aligning encoders with a chosen set of functions. We formalize affine alignment between encoders and show that it provides upper bounds on the differences in performance on downstream tasks. Experiments show our notion of intrinsic affine homotopy to be consistently predictive of downstream task behavior while revealing an asymmetric order in the space of encoders.

## Broader Impact

This paper presents foundational research about the similarity of language encoders. To the best of our knowledge, there are no ethical or negative societal implications to this work.

## Acknowledgements

Ryan Cotterell acknowledges support from the Swiss National Science Foundation (SNSF) as part of the "The Forgotten Role of Inductive Bias in Interpretability" project. Anej Svete is supported by the ETH AI Center Doctoral Fellowship. Robin Chan acknowledges support from FYAYC. We thank Raphaël Baur and Furui Cheng for helpful discussions and reviews of the current manuscript.

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

# A Notation

| Symbol | Meaning | Introduced |
|---|---|---|
| $D$ | Size of string representations. | §1 |
| $V$ | $D$-dimensional $\mathbb{R}$-vector space. | §2 |
| $\overline{\mathbb{R}}_+$ | $\mathbb{R}_+ \cup \{\infty\}$ | Def. 3.1 |
| $[N] \subset \mathbb{N}$ | The set $\{1, \ldots, N\}$ for $N \in \mathbb{N}$. | §5 |
| $\Delta^{N-1}$ | The $N-1$-dimensional probability simplex. | §1 |
| $\mathcal{E}_V$ | The vector space of language encoders. | §2 |
| $\mathcal{E}_b$ | The subspace of bounded language encoders. | §2 |
| $\mathrm{Aff}(V)$ | The set of (invertible) affine transformations on $V$. | §3.2 |
| $\mathrm{GL}(V)$ | The group of all invertible linear maps on $V$ to itself. | §3.2 |
| $\mathrm{O}(V)$ | The orthogonal group of $V$; the group of norm-preserving linear maps on $V$ | §3.3 |
| $d$ | A general (hemi-)metric. | Def. 3.2 |
| $d_\infty$ | The $\infty$ (uniform convergence) norm on $\mathcal{E}_V$. | §3.2 |
| $d^{\mathcal{H}}$ | The Hausdorff–Hoare map of $d$. | Def. 3.3 |
| $d^{\mathcal{H}, \mathrm{sym}}$ | The symmetrized Hausdorff–Hoare map. | Def. 3.3 |
| $d_S^{\mathcal{H}}$ | The Hausdorff–Hoare map where the sets in the arguments are computed by applying all transformations $s \in S$ to the two input elements. | Eq. (8) |
| $d_S$ | One-sided affine alignment measure over $S$. | Eq. (8a) |
| $\|\cdot\|_S$ | The $S$-homotopy norm of an encoder. | Eq. (8a) |
| $\gtrsim$ | A preorder. | §4 & §5 |
| $\simeq$ | An equivalence relation. | §4 |

Table 2: A summary of notation used in the paper.

# B Limitations

In this section, we address some of our work's limitations.

**Non-Linear Encoder Relationships.** This work focuses on affine similarity between encoders. As we find and discuss in §4 with the example of MULTIBERTs, language encoder representations may generally not be exactly affinely related. Nevertheless, understanding the affine-homotopy relationships on $\mathcal{E}_V$ still helps us to make conclusions about practical findings as in §7.

**Linear Classifiers.** Our work provides precise theoretical guarantees on the performance of linear classifiers applied to affinely-related encoders. In practice, task fine-tuning can take the form of more complex models, such as re-training entire pre-trained models. This work does not cover such more complex fine-tuning techniques.

**Numerical Approximations.** To bridge our theoretical findings on affine homotopy relationships in $\mathcal{E}_V$ with their practical implementations in §7, we concede several approximations. For instance, while $d^{\mathcal{H}}_{\mathcal{V}(V,\Delta)}$ is valuable in analysis, optimizing Eq. (23) directly is computationally challenging and requires costly approximations. Similarly, in computing intrinsic distances across all representation layers in Fig. 1, we optimize for mean squared error (MSE) and evaluate the maximum loss instead of optimizing for it directly, which serves as an approximation of $d$ that results in more stable optimization given computational constraints. Finally, we address numerical inaccuracies encountered during singular value decomposition (SVD) computations in §7, which we mitigate by tuning the rank according to precision $\epsilon$.

# C Additional Related Work

In this section, we complement our discussion in §6 and §8 with additional related work.

**Representational and Functional Similarity.** Our work is related to the ongoing efforts to quantify the similarity between neural networks. Much related work discusses similarity measures in terms of the invariance properties of neural networks [12, 22, *inter alia*]; see Klabunde et al. [21] for a recent comprehensive survey. Notably, Klabunde et al. [21] compile various *representational* [19, 23, 27, 32, 39, *inter alia*] and *functional* ways to measure similarity, which are related to our notions of intrinsic and extrinsic homotopy, respectively. Whereas our notion of intrinsic affine homotopy fits into the class of linear alignment-based measures [12, 27, 42, *inter alia*] as described in §6, the notion of extrinsic similarity fits into the broader category of performance-based functional measures [1, 8, 26]. Most relevantly, Boix-Adsera et al. [3] propose the GULP metric that provides a bound on the expected prediction dissimilarity for norm-one-bounded ridge regression.

**Similarity Measures as Metrics.** A line of work draws from *statistical shape analysis* [37] to motivate the development of similarity measures that are that conform to axioms of valid metrics [3, 36, 42]. Learning within proper metric spaces provides certain theoretical guarantees [2, 5, 10, 41, 43]. For example, Williams et al. [42] derive two families of *generalized shape metrics*, modifying existing dissimilarity measures to ensure they meet metric criteria. Notably, one of these generalized shape metrics is based on linear regression over the group of linear isometries, similar to the approach derived for encoder maps in Prop. 3.1.

**Understanding Similarity of Language Encoders.** Finally, several previous works characterize the landscape of language encoders and their sensitivity to slight changes to the pre-training or fine-tuning procedure [9, 13, 44]. This prompted multi-seed releases of encoders such as BERT [35, 44] that are frequently used for robustness or sensitivity analysis [12, 33], similar to the one presented in this work.

# D   Addenda on Affine Homotopy

In this section, we provide additional derivations and proofs complementing the discussion in §3–§5.

## D.1   Preliminaries on Hemi-Metric Spaces

**Definition D.1.** *Let $(X, d)$ be a hemi-metric space. The **open ball** $B(x, \epsilon)$ of center $x$ and radius $\epsilon > 0$, is the set $\{y \in X \mid d(x, y) < \epsilon\}$. The open balls form a base for the open ball topology.*[26]

**Lemma 4.1.** *Let $(X, d)$ be a hemi-metric space. The relation $(x \gtrsim_d y$ iff $d(x, y) = 0)$ is a **preorder**[27] and it will be called the **specialization ordering** of $d$.*

**Example D.1.** An example of a specialization ordering is the prefix ordering of strings $\leqslant_{\text{prefix}}$[28]. More precisely, for any $\boldsymbol{y}, \boldsymbol{y}' \in \Sigma^*$, we define $d_{\Sigma^*}(\boldsymbol{y}, \boldsymbol{y}')$ to be zero if $\boldsymbol{y}$ is a prefix of $\boldsymbol{y}'$ and $2^{-n}$ otherwise, where $n$ is the length of the longest prefix of $\boldsymbol{y}$ that is also a prefix of $\boldsymbol{y}'$. Then $(\Sigma^*, d_{\Sigma^*})$ is a hemi-metric space whose specialization ordering is $\leqslant_{\text{prefix}}$.                                                              //

**Lemma D.1.** *Let $(X, d)$ be a hemi-metric space.*

1.  *The set $\{x \in X \mid d^{\mathcal{H}}(x, E) = 0\}$ is exactly the closure of $E$ in the open ball topology.*

2.  *For any $x, x' \in X$, we have the inequality $d^{\mathcal{H}}(x, E) \leqslant d(x, x') + d^{\mathcal{H}}(x', E)$. If $d$ is a metric, then $d^{\mathcal{H}}(\cdot, E)$ is 1-Lipschitz from $(X, d)$ to $\overline{\mathbb{R}}_+$.*

3.  *Let $\mathcal{Z} \subset \mathcal{P}(X)$ be any space of non-empty subsets of $X$. The **Hausdorff–Hoare map** $d^{\mathcal{H}}$ is hemi-metric on $\mathcal{Z}$. Its specialization ordering $\gtrsim_{d^{\mathcal{H}}}$ is given by $E \gtrsim_{d^{\mathcal{H}}} E'$ iff[29] $E \subset cl(E')$, iff $cl(E) \subset cl(E')$.*

*Proof.* See Goubault-Larrecq [17, Lemma 6.1.11, Proposition 6.2.16 & Lemma 7.5.1].                ∎

---

[26]This, by definition, is the topology generated by all open balls.

[27]A reflexive and transitive relation on $X$.

[28]Defined by $\boldsymbol{y} \leqslant_{\text{prefix}} \boldsymbol{y}'$ if $\boldsymbol{y}$ is a prefix of $\boldsymbol{y}'$.

[29]Here, the closure is with respect to the topology defined by $d$.

## D.2 Additional Derivations: Affine Alignment Measures

**Remark 3.2.** $d_{\mathrm{Aff}(V)}$ *defined in Eq.* (8a) *is not a metric on* $\mathcal{E}_V$.[30] *Further, when* $S = \mathrm{Aff}(V)$, *the map* $\inf_{\psi, \psi' \in \mathrm{Aff}(V)} \|\psi \circ \boldsymbol{h} - \psi' \circ \boldsymbol{g}\|_\infty$ *is trivially zero by Cor. D.1.*

*Proof.* To see that $d_{\mathrm{Aff}(V)}$ is not a metric, consider the following two encoders: $\boldsymbol{g}(\boldsymbol{y}) = |\boldsymbol{y}| \cdot e$, where $e \in V$ is any fixed vector, and $\boldsymbol{h}$ be any map from $\Sigma^*$ to the ball $B(0_V, 1)$ of radius one. In such a case, we have $d_{\mathrm{Aff}(V)}(\boldsymbol{h}, \boldsymbol{g}) = \infty$. Even on the space of bounded encoders [31] $d_{\mathrm{Aff}(V)}$ is not a metric. We provide the following counter-example: Let $\boldsymbol{h}$ be any rank $R$ encoder, e.g., $\boldsymbol{h}$ can be any map that sends the first $R$ strings to the basis of $V$. Let $A$ be a non-invertible linear map of $V$ and set $\boldsymbol{g} = A(\boldsymbol{h})$. Then clearly $d_{\mathrm{Aff}(V)}(\boldsymbol{h}, \boldsymbol{g}) = 0$, but $d_{\mathrm{Aff}(V)}(\boldsymbol{g}, \boldsymbol{h})$ can not be zero for dimensionality reasons (see Thm. 4.1). ∎

**Lemma D.2** (Hausdorff Distance). *Let* $E, E' \subset \mathcal{E}_V$. *The map*

$$d_\infty^{\mathcal{H}, \, sym}(E, E') \overset{\text{def}}{=} \max(d_\infty^{\mathcal{H}}(E, E'), d_\infty^{\mathcal{H}}(E', E)) = \sup_{\boldsymbol{h} \in \mathcal{E}_V} |d_\infty^{\mathcal{H}}(\boldsymbol{h}, E) - d_\infty^{\mathcal{H}}(\boldsymbol{h}, E')| \tag{24}$$

*is an extended pseudo-metric on* $\mathcal{P}(\mathcal{E}_V) \backslash \{\varnothing\}$.

*Proof.* It follows readily from Lem. D.1. See also Burago et al. [4, §7.3.1]. ∎

For any affine subgroup $S \subset \mathrm{Aff}(V)$, let $S(\boldsymbol{h}) \overset{\text{def}}{=} \{\psi(\boldsymbol{h}) \mid \psi \in S\}$. It then follows immediately from Lem. D.2 that the map $d_S^{\mathcal{H}, \, sym}(\boldsymbol{h}, \boldsymbol{g}) \overset{\text{def}}{=} d_\infty^{\mathcal{H}, \, sym}(S(\boldsymbol{h}), S(\boldsymbol{g}))$ is an extended pseudo-metric on $\mathcal{E}_V$.

**Lemma D.3.** *For any* $\boldsymbol{h}, \boldsymbol{g} \in \mathcal{E}_V$, *any* $\psi \in \mathrm{Iso}(V)$ *and any non-empty* $S \subset \mathcal{E}_V$, *we have*

$$d_S(\psi \circ \boldsymbol{h}, \boldsymbol{g}) = d_{\psi^{-1}S}(\boldsymbol{h}, \boldsymbol{g}). \tag{25}$$

*In particular,* $d_{\mathrm{Iso}(V)}(\psi \circ \boldsymbol{h}, \boldsymbol{g}) = d_{\mathrm{Iso}(V)}(\boldsymbol{h}, \boldsymbol{g})$.

*Proof.* Lem. D.3 follows by definition $d_\infty(\psi \circ \boldsymbol{h}, \psi \circ \boldsymbol{g}) = d_\infty(\boldsymbol{h}, \psi^{-1} \circ \psi \circ \boldsymbol{g})$. ∎

**Proposition 3.1.** *The pair* $(\mathcal{E}_V, d_{\mathrm{Iso}(V)})$ *is an extended pseudo-metric space.*

*Proof.* Using Lem. D.3, one can show that $d_{\mathrm{Iso}(V)}(\boldsymbol{h}, \boldsymbol{g}) = d_\infty^{\mathcal{H}, \, sym}(\mathrm{Iso}(\boldsymbol{h}), \mathrm{Iso}(\boldsymbol{g}))$, where $\mathrm{Iso}(\boldsymbol{h}) \overset{\text{def}}{=} \{\psi \circ \boldsymbol{h} : \psi \in \mathrm{Iso}(V)\}$. The proposition follows then from Lem. D.2. ∎

For any $\psi \in \mathrm{Aff}(V)$ and any $\boldsymbol{h} \in \mathcal{E}_V$, we then have

$$\psi \circ \boldsymbol{h} \in \mathcal{E}_b \Leftrightarrow \boldsymbol{h} \in \mathcal{E}_b \Leftrightarrow \|\boldsymbol{h}\|_\infty < \infty.$$

**Lemma D.4.**

1. *If* $\boldsymbol{h} \in \mathcal{E}_b$, *then*
$$\|\boldsymbol{h}\|_{\mathrm{Iso}(V)} = \|\boldsymbol{h}\|_{\mathcal{T}} = r_{\boldsymbol{h}},$$
   *where* $r_{\boldsymbol{h}}$ *denotes the radius of* $\boldsymbol{h}$, *which we define as the radius of the minimum enclosing ball of the set* $\boldsymbol{h}(\Sigma^*)$, *and the* $\| \cdot \|_{\mathrm{Iso}(V)}$ *norm is defined as in Eq.* (8).

2. *For any* $\psi \in \mathrm{Aff}(V)$ *and a subset* $S \subset \mathrm{Aff}(V)$ *normalized[32] by* $\psi$ *and containing* $\mathcal{T}$. *Then*
$$\|\psi \circ \boldsymbol{h}\|_S \leqslant \|\psi_{lin}\|_V \cdot \|\boldsymbol{h}\|_S,$$
   *where the* $\| \cdot \|_S$ *norm is defined as in Eq.* (8).

*Proof.*

---

[30] See App. D.2 for a derivation.

[31] Recall $\mathcal{E}_b \overset{\text{def}}{=} \{\boldsymbol{h} \in \mathcal{E}_V \mid \boldsymbol{h}(\Sigma^*) \text{ is bounded}\}$.

[32] The set $S$ is normalized by $\psi$ if $\psi^{-1} \circ \phi \circ \psi \in S$ for all $\phi \in S$.

1. Let $t \in \mathcal{T}$ be the translation moving the center of the ball enclosing $\boldsymbol{h}(\Sigma^*)$ to the center $0_V$. Hence

$$\|\boldsymbol{h}\|_{\mathrm{Iso}(V)} \leqslant \|\boldsymbol{h}\|_{\mathcal{T}} \leqslant \|t \circ \boldsymbol{h}\|_\infty = r_{\boldsymbol{h}}$$

Now observe that for any other isometry $\psi \neq t$, then $r_{\psi \circ \boldsymbol{h}} = r_{\boldsymbol{h}}$. The ball $B(0_V, \|\psi \circ \boldsymbol{h}\|_\infty)$ clearly contains all points in $\psi \circ \boldsymbol{h}(\Sigma^*)$, hence by definition of the radius $r_{\psi \circ \boldsymbol{h}}$ we must have $\|\psi \circ \boldsymbol{h}\|_\infty \leqslant r_{\boldsymbol{h}}$, which finishes the proof of 1.

2. Write $\psi = \phi_{\mathrm{lin}} \circ t$, with $t \in \mathcal{T}$. We then have

$$\|\psi \circ \boldsymbol{h}\|_S = \inf_{\phi \in S} \|\phi(\psi \circ \boldsymbol{h})\|_\infty$$

$$= \inf_{\phi \in S} \|\psi_{\mathrm{lin}}(\underbrace{\psi_{\mathrm{lin}}^{-1} \circ \phi \circ \psi_{\mathrm{lin}} \circ t}_{\in S} \circ \boldsymbol{h})\|_\infty$$

Note that $\phi \mapsto \psi_{\mathrm{lin}}^{-1} \circ \phi \circ \psi_{\mathrm{lin}} \circ t$ is by definition a bijection of $S$, hence

$$\|\psi \circ \boldsymbol{h}\|_S = \inf_{\phi \in S} \|\psi_{\mathrm{lin}}(\phi \circ \boldsymbol{h})\|_\infty$$

$$= \inf_{\phi \in S} \sup_{\boldsymbol{y} \in \Sigma^*} |\psi_{\mathrm{lin}}((\phi \circ \boldsymbol{h})(\boldsymbol{y}))|_V$$

$$\leqslant \|\psi_{\mathrm{lin}}\|_V \inf_{\phi \in S} \sup_{\boldsymbol{y} \in \Sigma^*} |(\phi \circ \boldsymbol{h})(\boldsymbol{y})|_V$$

$$= \|\psi_{\mathrm{lin}}\|_V \cdot \|\boldsymbol{h}\|_S. \qquad \blacksquare$$

**Corollary D.1.** Let $S \supset \mathcal{T}$ such that $\inf_{\psi \in S} \|\psi_{\mathrm{lin}}\|_V$.[33] Then, $d_S(\boldsymbol{h}, \boldsymbol{g}) \overset{\mathrm{def}}{=} \inf_{\psi, \psi' \in S} \|\psi \circ \boldsymbol{h} - \psi' \circ \boldsymbol{g}\|_\infty = 0$ for all $\boldsymbol{h}, \boldsymbol{g} \in \mathcal{E}_b$.

*Proof.* Note that $d_S(\psi \circ \boldsymbol{h}, \boldsymbol{g}) \leqslant \|\psi_{\mathrm{lin}}\|_V \cdot d_S(\boldsymbol{h}, \boldsymbol{g})$, which follows from Lem. D.4. Hence

$$d_S(\boldsymbol{h}, \boldsymbol{g}) = \inf_{\psi \in S} d_S(\psi \circ \boldsymbol{h}, \boldsymbol{g})$$

$$\leqslant \underbrace{\inf_{\psi \in S} \|\psi_{\mathrm{lin}}\|_V}_{=0} d_S(\boldsymbol{h}, \boldsymbol{g}). \qquad \blacksquare$$

## D.3 Proofs: Intrinsic Affine Homotopy

**Lemma 4.2.** *The relation $\gtrsim_{\mathrm{Aff}}$ is a preorder on $\mathcal{E}_V$.*

*Proof.* Since $d_{\mathrm{Aff}(V)}(\psi \circ \boldsymbol{h}, \boldsymbol{g}) \leqslant \|\psi_{\mathrm{lin}}\|_V \cdot d_{\mathrm{Aff}(V)}(\boldsymbol{h}, \boldsymbol{g})$ (see Lem. D.4),

$$d_{\mathrm{Aff}(V)}(\boldsymbol{h}, \boldsymbol{g}) = 0 \Leftrightarrow d_{\mathrm{Aff}(V)}^{\mathcal{H}}(\boldsymbol{h}, \boldsymbol{g}) = 0.$$

Therefore, the relation $\gtrsim_{\mathrm{Aff}}$ is the specialization ordering of the hemi-metric $d_{\mathrm{Aff}(V)}^{\mathcal{H}}$. $\qquad \blacksquare$

**Theorem 4.1.** *For $\boldsymbol{h}, \boldsymbol{g} \in \mathcal{E}_V$, we have*

$$\boldsymbol{h} \gtrsim_{\mathrm{Aff}} \boldsymbol{g} \Leftrightarrow \boldsymbol{h} = \psi(\pi_{\boldsymbol{h}} \circ \boldsymbol{g}) \text{ for some } \psi \in \mathrm{Aff}(V) \tag{10}$$

*where, $\pi_{\boldsymbol{h}}$ is the orthogonal projection of $V$ onto $V_{\boldsymbol{h}}$. In particular, if $d_{\mathrm{Aff}(V)}(\boldsymbol{h}, \boldsymbol{g}) = 0$ then $\mathrm{rank}(\boldsymbol{h}) \leqslant \mathrm{rank}(\boldsymbol{g})$. If in addtion, we know $\mathrm{rank}(\boldsymbol{g}) = \mathrm{rank}(\boldsymbol{h})$, then $\boldsymbol{g}$ must by an affine transformation of $\boldsymbol{g}$, i.e., $\boldsymbol{h} = \psi \circ \boldsymbol{g}$ for some $\psi \in \mathrm{Aff}(V)$.*

*Proof.*

1. Recall from §3.2 that $\mathcal{E}_V$ is complete with respect to the metric $d_\infty$. The condition $d(\boldsymbol{h}, \boldsymbol{g})_{\mathrm{Aff}(V)} = 0$ simply means that there exists $\phi_n \in \mathrm{Aff}(V)$ such that $\lim_{n \to \infty} \phi_n \circ \boldsymbol{g} = \boldsymbol{h}$ in $\mathcal{E}_V$, in other words $\boldsymbol{h} \in \overline{\mathrm{Aff}(\boldsymbol{g})}$, i.e., $\boldsymbol{h}$ lies in the closure of $\mathrm{Aff}(\boldsymbol{g})$ in $\mathcal{E}_V$.

---

[33]This is, for example, the case if $S$ is a group and there exists $\phi \in S$ such that $\|\phi_{\mathrm{lin}}\|_V < 1$, e.g., $S = \mathrm{Aff}(F)$.

Let $\mathcal{B}_{\boldsymbol{h}} \subset \Sigma^*$ such that $\boldsymbol{h}(\mathcal{B}_{\boldsymbol{h}})$ is a basis for $V_{\boldsymbol{h}}$. Therefore, there exists $\epsilon > 0$ such that any family[34]

$$(v_{\boldsymbol{y}})_{\boldsymbol{y}} \in \prod_{\boldsymbol{y} \in \mathcal{B}_{\boldsymbol{h}}} B(\boldsymbol{h}(\boldsymbol{y}), \epsilon)$$

has rank $\dim_{\mathbb{R}}(V_{\boldsymbol{h}})$. This shows that there exists $N \geqslant 1$ such that for any $n \geqslant N$ one has $\|\boldsymbol{h} - \phi_n \circ \boldsymbol{g}\|_\infty < \epsilon$, and

$$\mathrm{rank}(\{\phi_n \circ \boldsymbol{g}(\boldsymbol{y}) \colon \boldsymbol{y} \in \mathcal{B}_{\boldsymbol{h}}\}) = \mathrm{rank}(\boldsymbol{h}).$$

Which implies in particular

$$\dim_{\mathbb{R}}(V_{\boldsymbol{h}}) \leqslant \dim_{\mathbb{R}}(V_{\boldsymbol{g}}) \text{ i.e., } \mathrm{rank}\,\boldsymbol{h} \leqslant \mathrm{rank}\,\boldsymbol{g}. \tag{26}$$

2. If $\mathrm{rank}(\boldsymbol{g}) = \mathrm{rank}(\boldsymbol{h}) = D$, then $\lim_{n\to\infty} \phi_n = \phi$, where $\phi$ is the affine map given by $\boldsymbol{g}(\boldsymbol{y}) \mapsto \boldsymbol{h}(\boldsymbol{y})$ for $\boldsymbol{y} \in \mathcal{B}_{\boldsymbol{h}}$. Indeed, for any $v = \sum_{b \in \mathcal{B}_{\boldsymbol{h}}} \lambda_b b \in V$, we have

$$\|(\phi - \phi_n)(v)\| \leqslant \|\boldsymbol{h} - \psi_n \circ \boldsymbol{g}\|_\infty \sum_{b \in \boldsymbol{h}(\mathcal{B}_{\boldsymbol{h}})} |\lambda_b| \leqslant c\|\boldsymbol{h} - \psi_n \circ \boldsymbol{g}\|_\infty \|v\|_V$$

for some constant $c > 0$, since all norms on $V$ are equivalent. Hence, $\lim_{n\to\infty} \|\phi - \phi_n\|_V = 0$, which shows the claim. Accordingly, we must have $\phi \circ \boldsymbol{g} = \boldsymbol{h}$.

Now we can prove Eq. (10):

3 $\Rightarrow$. Given that $\|\boldsymbol{h} - \pi_{\boldsymbol{h}} \circ \phi_n \circ \boldsymbol{g}\|_\infty \leqslant \|\boldsymbol{h} - \phi_n \circ \boldsymbol{g}\|_\infty$, we also have $\lim_{n\to\infty} \pi_{\boldsymbol{h}} \circ \phi_n \circ \boldsymbol{g} = \boldsymbol{h}$.

Write $\pi^\perp$ for the orthogonal projection on $V^\perp$ and set $\pi_{\boldsymbol{h},n} = \pi_{\boldsymbol{h}} \oplus \frac{1}{n\|\phi_n\|}\pi_{\boldsymbol{h}}^\perp$. Note that $\lim_{n\to\infty} \pi_{\boldsymbol{h},n} = \pi_{\boldsymbol{h}}$. Accordingly,

$$\lim_{n\to\infty} \psi_n(\pi \circ \boldsymbol{g}) = \boldsymbol{h},$$

where $\psi_n = \pi_{\boldsymbol{h},n}\phi_n\pi_{\boldsymbol{h},n}^{-1}$. From this, we deduce that

$$d_{\mathrm{Aff}(V)}(\boldsymbol{h}, \pi_{\boldsymbol{h}} \circ \boldsymbol{g}) = 0.$$

Now applying 2. yields $\boldsymbol{h} = \phi(\pi_{\boldsymbol{h}} \circ \boldsymbol{g})$ for some $\phi_h \in \mathrm{Aff}(V_{\boldsymbol{h}})$, or $\boldsymbol{h} = \phi(\pi_{\boldsymbol{h}} \circ \boldsymbol{g})$ where $\phi = \phi_h \oplus \pi_{\boldsymbol{h}}^\perp \in \mathrm{Aff}(V)$.

3 $\Leftarrow$. Assume now that $\boldsymbol{h} = \phi(\pi_{\boldsymbol{h}} \circ \boldsymbol{g})$ for some $\phi \in \mathrm{Aff}(V)$. Then $\boldsymbol{h} = \lim_{n\to\infty} \phi \circ \pi_{\boldsymbol{h},n}(\boldsymbol{g})$, where $\pi_{\boldsymbol{h},n} = \pi_{\boldsymbol{h}} \oplus \frac{1}{n}\pi_{\boldsymbol{h}}^\perp$, which shows the desired implication. $\blacksquare$

### D.4 Proofs: Extrinsic Homotopy

**Lemma 5.2.** *Let $\boldsymbol{h}, \boldsymbol{g} \in \mathcal{E}_V$. We have*

1. *There exists a constant $c(\lambda) > 0$ such that for any $\psi \in \mathrm{Aff}(V, W)$*

$$d_{\infty, \Delta^{N-1}}^{\mathcal{H}}(\mathrm{softmax}_\lambda(\psi \circ \boldsymbol{h}), \mathcal{V}_N(\boldsymbol{g})) \leqslant c(\lambda)\|\psi_{lin}\| d_{\mathrm{Aff}(V)}(\boldsymbol{h}, \boldsymbol{g}).$$

2. $d_{\mathcal{V}(V,\Delta)}^{\mathcal{H}}(\boldsymbol{h}, \boldsymbol{g}) \leqslant c(\lambda) d_{\mathrm{Aff}(V,W)}^{\mathcal{H}}(\boldsymbol{h}, \boldsymbol{g}).$

3. $d_{\mathrm{Aff}(V)}(\boldsymbol{h}, \boldsymbol{g}) = 0 \Rightarrow d_{\mathrm{Aff}(V,W)}^{\mathcal{H}}(\boldsymbol{h}, \boldsymbol{g}) = 0 \Rightarrow d_{\mathcal{V}(V,\Delta)}^{\mathcal{H}}(\boldsymbol{h}, \boldsymbol{g}) = 0.$

*Proof.*

1. Clearly,

$$
\begin{aligned}
d_{\infty, \Delta^{N-1}}^{\mathcal{H}}(\mathrm{softmax}_\lambda(\psi \circ \boldsymbol{h}), \mathcal{V}_N(\boldsymbol{g})) &\leqslant c(\lambda) d_{\mathcal{V}(V,W)}(\psi \circ \boldsymbol{h}, \mathrm{Aff}_{V,W}(\boldsymbol{g})) \\
&\leqslant c(\lambda) \inf_{\psi' \in \psi \circ \mathrm{Aff}(V)} \|\psi \circ \boldsymbol{h} - \psi' \circ \boldsymbol{g}\|_{\infty, W} \\
&= c(\lambda) \inf_{\psi' \in \mathrm{Aff}(V)} \|\psi(\boldsymbol{h} - \psi' \circ \boldsymbol{g})\|_{\infty, W} \\
&= c(\lambda)\|\psi_{\mathrm{lin}}\| d_{\mathrm{Aff}(V)}(\boldsymbol{h}, \boldsymbol{g}).
\end{aligned}
$$

---

[34]close enough to $\boldsymbol{h}(\mathcal{B}_{\boldsymbol{h}})$.

where, the first inequality follows from the fact that $\mathrm{softmax}_\lambda$ is $c(\lambda)$-Lipschitz for some constant that depends on $\lambda$ [16, Proposition 4].

2. & 3. are are immediate consequences of 1. ∎

**Theorem 5.1** ($\epsilon$-Intrinsic $\Rightarrow \mathcal{O}(\epsilon)$-Extrinsic)**.** *Let $\boldsymbol{h}, \boldsymbol{g} \in \mathcal{E}_V$ be two encoders. Then,*

$$d^{\mathcal{H}}_{\mathcal{V}(V,\Delta)}(\boldsymbol{h}, \boldsymbol{g}) \leqslant c(\lambda)\, d^{\mathcal{H}}_{\mathrm{Aff}(V)}(\boldsymbol{h}, \boldsymbol{g}).$$

*Proof.* Let $\psi \in \mathrm{Aff}(V, W)$. There exists a linear map $A\colon V \to W$ and a $\phi_V \in \mathrm{GL}(V)$, such that $\psi = A \circ \phi$ and $\|A\| = 1$. Accordingly, Lem. 5.2 yields

$$d^{\mathcal{H}}_{\infty, \Delta^{N-1}}(\mathrm{softmax}_\lambda(\psi \circ \boldsymbol{h}), \mathcal{V}_N(\boldsymbol{g})) \leqslant c(\lambda)\, d_{\infty, W}(\psi \circ \boldsymbol{h}, \mathrm{Aff}_{V,W}(\boldsymbol{g})) \tag{28a}$$

$$\leqslant c(\lambda) d_{\mathrm{Aff}(V)}(\phi_V \circ \boldsymbol{h}, \boldsymbol{g}) \tag{28b}$$

$$\leqslant c(\lambda) \sup_{\psi_V \in \mathrm{Aff}(V)} (d_{\mathrm{Aff}(V)}(\psi_V \circ \boldsymbol{h}, \boldsymbol{g})) \tag{28c}$$

$$= c(\lambda)\, d^{\mathcal{H}}_{\mathrm{Aff}(V)}(\boldsymbol{h}, \boldsymbol{g}). \tag{28d}$$

Therefore $d^{\mathcal{H}}_{\mathcal{V}(V,\Delta)}(\boldsymbol{h}, \boldsymbol{g}) \leqslant c(\lambda)\, d^{\mathcal{H}}_{\mathrm{Aff}(V)}(\boldsymbol{h}, \boldsymbol{g})$. ∎

# E  Addenda on Linear Alignment Methods for Finite Representation Sets

**Linear Regression**  A common way to evaluate the similarity of two representation matrices $\mathbf{H} \in \mathbb{R}^{N \times D}$ and $\mathbf{G} \in \mathbb{R}^{N \times D}$ is through linear regression. Linear regression finds the matrix $\hat{\mathbf{A}} \in \mathbb{R}^{D \times D}$ that minimizes the least squares error:

$$\hat{\mathbf{A}} = \operatorname*{argmin}_{\mathbf{A} \in \mathbb{R}^{D \times D}} \|\mathbf{G} - \mathbf{H}\mathbf{A}\|_F^2 = (\mathbf{H}^\top \mathbf{H})^{-1} \mathbf{H}^\top \mathbf{G}. \tag{29}$$

Let $\mathbf{H} = \mathbf{Q_H}\mathbf{R_H}$ and $\mathbf{G} = \mathbf{Q_G}\mathbf{R_G}$ be the QR-decomposition of $\mathbf{H}$ and $\mathbf{G}$, respectively. The goodness of fit is commonly evaluated through the R-squared value $R^2_{LR}$, i.e., as the proportion of variance in $\mathbf{G}$ explained by the fit:

$$R^2_{LR} = 1 - \frac{\|\mathbf{G} - \mathbf{H}\hat{\mathbf{A}}\|_F^2}{\|\mathbf{G}\|_F^2} = \frac{\|\mathbf{Q_G}^\top \mathbf{H}\|_F^2}{\|\mathbf{G}\|_F^2}. \tag{30}$$

To derive Eq. (30), consider the fitted value $\hat{\mathbf{G}}$

$$\hat{\mathbf{G}} = \mathbf{H}\hat{\mathbf{A}} = \mathbf{H}(\mathbf{H}^\top \mathbf{H})^{-1} \mathbf{H}^\top \mathbf{G} \tag{31a}$$

$$= \mathbf{Q_H}\mathbf{R_H}(\mathbf{R_H}^\top \mathbf{Q_H}^\top \mathbf{Q_H}\mathbf{R_H})^{-1} \mathbf{R_H}^\top \mathbf{Q_H}^\top \mathbf{G} \tag{31b}$$

$$= \mathbf{Q_H}\mathbf{Q_H}^\top \mathbf{G}. \tag{31c}$$

The residuals are therefore

$$\|\mathbf{G} - \hat{\mathbf{G}}\|_F^2 = \mathrm{tr}((\mathbf{G} - \hat{\mathbf{G}})^\top (\mathbf{G} - \hat{\mathbf{G}})) \tag{32a}$$

$$= \mathrm{tr}((\mathbf{G} - \hat{\mathbf{G}})^\top \mathbf{G}) \qquad (\text{32b, residuals orthogonal to fitted values})$$

$$= \mathrm{tr}(\mathbf{G}^\top \mathbf{G}) - \mathrm{tr}(\mathbf{G}^\top \mathbf{Q_H}\mathbf{Q_H}^\top \mathbf{G}) \tag{32c}$$

$$= \|\mathbf{G}\|_F^2 - \|\mathbf{Q_H}^\top \mathbf{G}\|_F^2. \tag{32d}$$

With this, we can compute the coefficient of determination as

$$R^2_{LR} = 1 - \frac{\|\mathbf{G} - \hat{\mathbf{G}}\|_F^2}{\|\mathbf{G}\|_F^2} = 1 - \frac{\|\mathbf{G}\|_F^2 - \|\mathbf{Q_H}^\top \mathbf{G}\|_F^2}{\|\mathbf{G}\|_F^2} = \frac{\|\mathbf{Q_H}^\top \mathbf{G}\|_F^2}{\|\mathbf{G}\|_F^2}. \tag{33}$$

**Orthogonal Procrustes Problem.** Let $\mathbf{G} \in \mathbb{R}^{N \times D}$ and $\mathbf{H} \in \mathbb{R}^{N \times D}$ representation matrices. In the orthogonal Procrustes problem, we seek to find the *orthogonal* matrix $\mathbf{A}$ that best maps $\mathbf{H}$ to $\mathbf{G}$:

$$\underset{\mathbf{A} \in O(V)}{\operatorname{argmin}} \|\mathbf{H} - \mathbf{AG}\|_F. \tag{34}$$

Since

$$\begin{aligned}
\|\mathbf{G} - \mathbf{HA}\|_F^2 &= \operatorname{tr}((\mathbf{G} - \mathbf{HA})^\top (\mathbf{G} - \mathbf{HA})) \\
&= \operatorname{tr}(\mathbf{G}^\top \mathbf{G}) - \operatorname{tr}(\mathbf{G}^\top \mathbf{HA}) - \operatorname{tr}(\mathbf{A}^\top \mathbf{H}^\top \mathbf{G}) + \operatorname{tr}(\mathbf{A}^\top \mathbf{H}^\top \mathbf{HA}) \\
&= \|\mathbf{G}\|_F^2 + \|\mathbf{H}\|_F^2 - 2\operatorname{tr}(\mathbf{A}^\top \mathbf{H}^\top \mathbf{G}),
\end{aligned}$$

an equivalent objective to Eq. (34) is

$$\hat{\mathbf{A}} = \underset{\mathbf{A} \in O(V)}{\operatorname{argmax}} \langle \mathbf{AH}, \mathbf{G} \rangle_F$$

Let $\mathbf{U}\boldsymbol{\Sigma}\mathbf{V}^\top$ be the singular-value decomposition of $\mathbf{H}^\top \mathbf{G}$, then

$$\hat{\mathbf{A}} = \underset{\mathbf{A} \in O(V)}{\operatorname{argmax}} \langle \mathbf{AH}, \mathbf{G} \rangle_F \tag{35a}$$

$$= \underset{\mathbf{A} \in O(V)}{\operatorname{argmax}} \langle \mathbf{A}, \mathbf{GH}^\top \rangle_F \tag{35b}$$

$$= \underset{\mathbf{A} \in O(V)}{\operatorname{argmax}} \langle \mathbf{A}, \mathbf{U}\boldsymbol{\Sigma}\mathbf{V}^\top \rangle_F \tag{35c}$$

$$= \underset{\mathbf{A} \in O(V)}{\operatorname{argmax}} \langle \mathbf{U}^\top \mathbf{AV}, \boldsymbol{\Sigma} \rangle_F \tag{35d}$$

where $\mathbf{U}^\top \mathbf{AV}$ is a product of orthogonal matrices, and, therefore, orthogonal. Since $\boldsymbol{\Sigma}$ is diagonal, Eq. (35d) is maximized by $\mathbf{U}^\top \hat{\mathbf{A}}\mathbf{V} = \mathbf{I}$, which means that $\hat{\mathbf{A}} = \mathbf{UV}^\top$.

**Canonical Correlation Analysis.** We can rewrite the CCA objective from Eq. (21) as

$$\max_{\mathbf{A}, \mathbf{B}} \operatorname{tr}(\mathbf{A}^\top \mathbf{HG}^\top \mathbf{B}) \quad \text{s.t.} \quad (\mathbf{A}^\top \mathbf{H})(\mathbf{A}^\top \mathbf{H})^\top = (\mathbf{B}^\top \mathbf{G})(\mathbf{B}^\top \mathbf{G})^\top = \mathbf{I}, \tag{36}$$

which, by definition of the Frobenius norm, is equivalent to Eq. (22). Let $\mathbf{M_{HG}} = \mathbf{HG}^\top$, $\mathbf{M_{HH}} = \mathbf{HH}^\top$, $\mathbf{M_{GG}} = \mathbf{GG}^\top$, and let $\mathbf{U}\boldsymbol{\Sigma}\mathbf{V}^\top = \mathbf{M_{HG}}$ be the singular-value decomposition of $\mathbf{M_{HG}}$. One can show that the optimum of Eq. (22) is found at $(\hat{\mathbf{A}}, \hat{\mathbf{B}}) = (\mathbf{M_{HH}}^{-\frac{1}{2}}\mathbf{U}, \mathbf{M_{GG}}^{-\frac{1}{2}}\mathbf{V})$. Because $\mathbf{A}^\top \mathbf{H}$, $\mathbf{B}^\top \mathbf{G}$, $\mathbf{U}$, and $\mathbf{V}$ are by definition orthogonal, we see that CCA first whitens the representations $(\mathbf{H}, \mathbf{G})$ through $(\mathbf{M_{HH}}^{-\frac{1}{2}}, \mathbf{M_{GG}}^{-\frac{1}{2}})$ and then orthogonally transforms them. This provides the intuition behind a close relationship between CCA and the Orthogonal Procrustes problem: For pre-whitened representation matrices, CCA (Eq. (22)) is equivalent to solving the Orthogonal Procrustes problem (Eq. (34)). To see this, let $\mathbf{W_H}$ and $\mathbf{W_G}$ be whitening transforms for $\mathbf{H}$ and $\mathbf{G}$, respectively. Then, Eq. (22) is equivalent to

$$\min_{\mathbf{A}, \mathbf{B} \in O(V)} \|\mathbf{A}^\top \mathbf{W_H}\mathbf{H} - \mathbf{B}^\top \mathbf{W_G}\mathbf{G}\|_F^2 \tag{37}$$

such that

$$(\mathbf{AW_H}\mathbf{H})(\mathbf{AW_H}\mathbf{H})^\top = \mathbf{AA}^\top = \mathbf{I}, \tag{38a}$$

$$(\mathbf{BW_G}\mathbf{G})(\mathbf{BW_G}\mathbf{G})^\top = \mathbf{BB}^\top = \mathbf{I}. \tag{38b}$$

Therefore, we can derive

$$\min_{\mathbf{A}, \mathbf{B} \in O(V)} \|\mathbf{A}^\top \mathbf{W_H}\mathbf{H} - \mathbf{B}^\top \mathbf{W_G}\mathbf{G}\|_F^2 = \min_{\mathbf{AB}^\top \in O(V)} \|\mathbf{A}^\top\| \|\mathbf{W_H}\mathbf{H} - \mathbf{AB}^\top \mathbf{W_G}\mathbf{G}\|_F^2$$

$$\text{(39a, } \mathbf{A} \in O(V))$$

$$= \min_{\mathbf{C} \in O(V)} \|\mathbf{W_H}\mathbf{H} - \mathbf{C}^\top \mathbf{W_G}\mathbf{G}\|.,$$

$$\text{(39b, } \mathbf{C} \stackrel{\text{def}}{=} \mathbf{AB}^\top \in O(V))$$

which is equivalent to solving the Orthogonal Procrustes problem (Eq. (34)) on the whitened matrices $\mathbf{W_H}\mathbf{H}$ and $\mathbf{W_G}\mathbf{G}$.

# F   Experimental Setup

In this section, we provide additional details about the setup and compute resources of the experiments in §7. To generate embeddings, we used the open-sourced code by Ren et al. [33]. Further, for Orthogonal Procrustes, CCA, PWCCA, and Linear CKA, we use the open source implementation by Ding et al. [12]. Our complete code is added as supplementary material.

**Models and Datasets.**   We first extract the $D = 768$ dimensional training set representations for SST-2, MRPC, RTE, CoLA, MNLI, and QQP across all 12 layers of ELECTRA [6], ROBERTA [28], and the 25 MULTIBERT [35] models from HuggingFace.[35] The models and the MRPC dataset are licensed under `Apache License 2.0`. The SST-2 dataset is licensed under the `Creative Commons CC0: Public Domain` license. The RTE dataset is licensed under the `CC BY 3.0` license. The CoLA dataset is licensed under the `CC BY-SA 4.0` license. The MNLI dataset is licensed under the `General Public License (GPL)`. THE QQP dataset is licensed under a custom non-commercial license.[36] The dataset statistics are shown in Tab. 3. We note that for all experiments, MNLI and QQP were shortened to the first 10K training samples due to computational limitations.

| Dataset | Task | Train Dataset Size | Domain |
|---------|------|--------------------|--------|
| SST-2 | Sentiment Analysis | 67K | Movie reviews |
| MRPC | Paraphrase Detection | 3.7K | News |
| RTE | Textual Entailment | 2.5K | Mixed |
| CoLA | Linguistic Acceptability | 8.5K | Miscellaneous |
| MNLI | Natural Language Inference | 393K | Multi-Genre |
| QQP | Paraphrase Detection | 364K | Social QA |

Table 3: Statistics for the used GLUE benchmark [40] datasets.

**Hyperparameters.**   Each experiment was run using RiemannSGD[37] as an optimizer as it initially produced the best convergence when computing our affine similarity measures. Further, to account for convergence artifacts, we ran the intrinsic similarity computation optimizations in each experiment for learning rates $[1\text{E-}4, 1\text{E-}3, 1\text{E-}2, 1\text{E-}1]$ and extrinsic computations for $[1\text{E-}3, 1\text{E-}2, 2\text{E-}2]$ and report the best result. When training the task-specific linear probing classifier $\psi'$ for $\hat{d}_{\psi'}$, we use the cross-entropy loss, RiemannSGD and optimize over the learning rates $[1\text{E-}2, 1\text{E-}1, 2\text{E-}1, 4\text{E-}1]$. For the computation of Hausdorff–Hoare map $d^{\mathcal{H}}$, we fixed a lr of $1\text{E-}3$ to save compute resources, as this lr generally leads to the best convergence in previous experiments. We used a batch size 64 and let optimization run for 20 epochs, keeping other parameters at default. For reproducibility, we set the initial seed to 42 during training.

**Generating Random Affine Maps.**   For the last experiment, we generate random affine maps. To approximate $d^{\mathcal{H}}$ we sample the matrix entries of the affine map from $\mathcal{N}(0, 1)$. We then additionally normalize the transformed representation matrix as this leads to better convergence. To approximate $\hat{d}_{\mathcal{V}(V,\Delta)}^{\mathcal{H}}$, we fit a linear probe on $\mathbf{H}$ to 100 sets of randomly generated class labels, for the embeddings of each task. The predictions of that probe then become what $\mathbf{G}$ affinely maps to. In both cases, the seeds are set ascendingly from 0.

**Compute Resources.**   We compute the embeddings on a single `A100-40GB` GPU, which took around two hours. All other experiments were run on `8-32` CPU cores, each with `8 GB` of memory. Computing extrinsic distances between 600 model combinations across both datasets usually takes 2-3 hours on 8 cores, whereas intrinsic computation is more costly, and can run up to 4 hours. Note our approximation of Hausdorff–Hoare maps (cf. Eq. (23)) across all models is significantly more costly due to our sampling approach and can take up to 72 hours to compute on 32 cores for large datasets such as SST-2, and up to 12 hours for MRPC. The resources needed for initially failed experiments do not significantly exceed the reported compute.

---

[35]https://huggingface.co/google
[36]https://www.quora.com/about/tos
[37]https://github.com/geoopt/geoopt

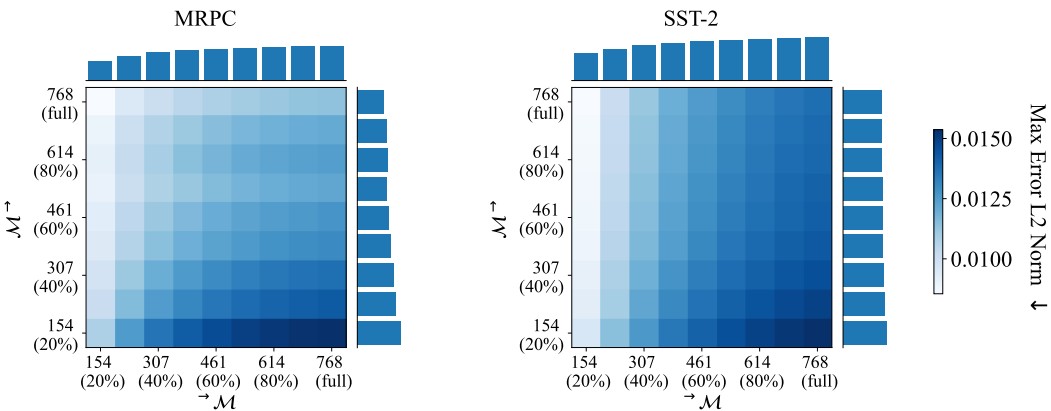

Figure 3: The effect of artificial rank deficiency averaged across MULTIBERTs. For each pair of embeddings $\mathbf{H}^{(i)}$ and $\mathbf{H}^{(j)}$ from MUTLIBERTs $\mathcal{M}^{(i)}$ and $\mathcal{M}^{(j)}$ we generate additional rank-deficient encoders $\mathbf{H}^{(i)}_{X\%}$ and $\mathbf{H}^{(j)}_{Y\%}$ with $X, Y \in \{20\%, ..., 90\%\}$ of the full rank through SVD truncation. We compute $d(\mathbf{H}^{(i)}_{Y\%}, \mathbf{H}^{(j)}_{X\%})$, for each pair of possible rank-deficiency and finally report the median across all MULTIBERTs on row $X$ and column $Y$ on the grid. We additionally show row-, and column medians.

# G    Additional Experimental Results

**The Influence of Encoder Rank Deficiency.**    In Thm. 4.1 we discuss how the relative rank of encoders influences their affine alignment and derive the equivalence relation $\simeq_{\mathrm{Aff}}$ conditioned on equal rank between encoders. To test the effect of unequal rank on affine alignment in an isolated setup, we artificially construct reduced-rank encoders through singular value decomposition (SVD) truncation. In Figure 3 we expectedly find a trend in how the encoder rank influences affine mappability. We additionally highlight that the computed distances are rather symmetric, with no clear differences when mapping *to* ($^{\rightarrow}\mathcal{M}$), rather than *from* ($\mathcal{M}^{\rightarrow}$) an encoder. Finally, we note the trend in the diagonal indicating that mapping between encoders of the same rank becomes easier between lower-rank encoders.

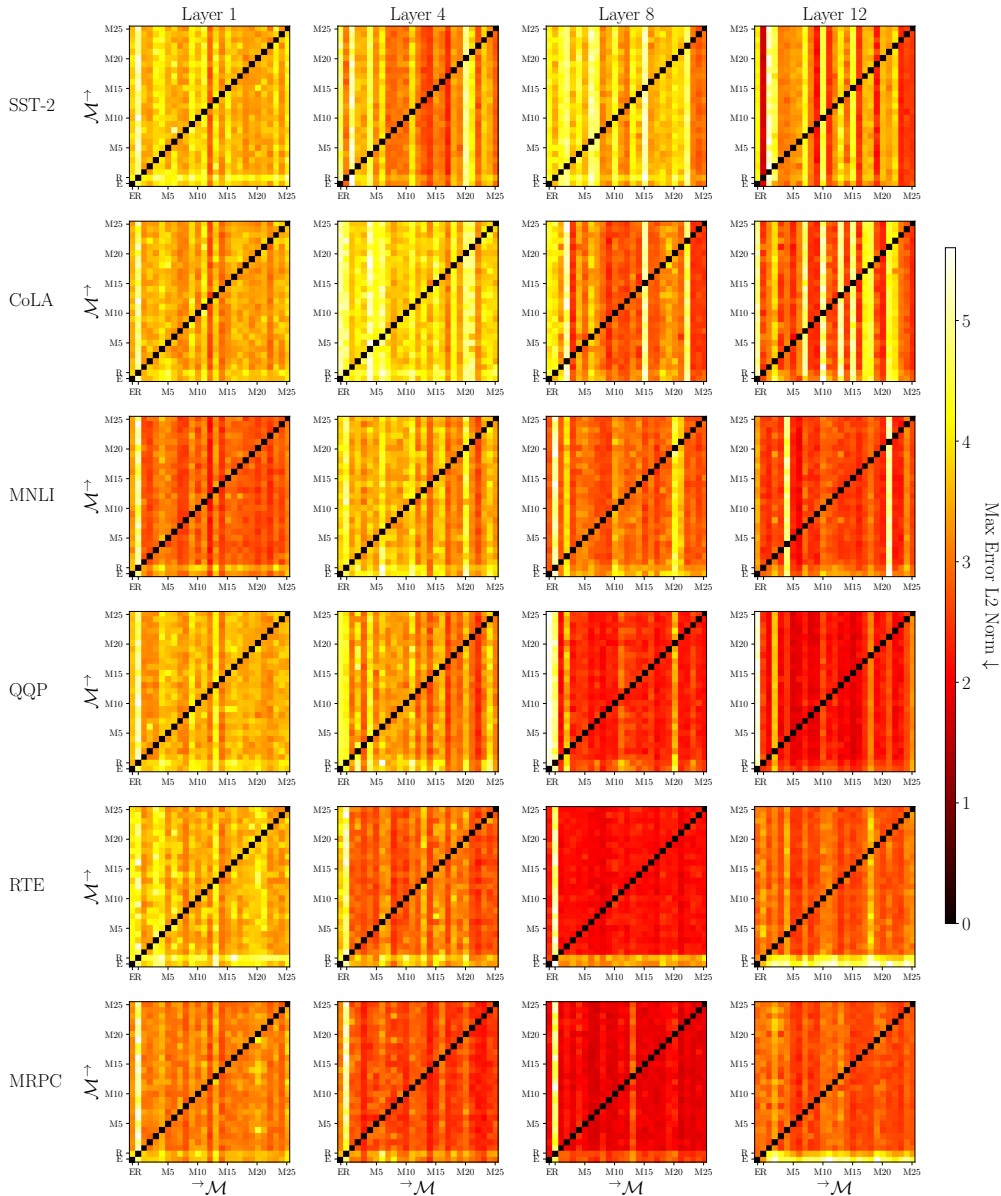

Figure 4: Asymmetry between ELECTRA (E), RoBERTa (R), and MULTIBERT encoders (M1-M25) across layers. For each pair of the encoders $\mathcal{M}^{(i)}$ and $\mathcal{M}^{(j)}$, we generate training set embeddings $\mathbf{H}^{(i)}, \mathbf{H}^{(j)} \in \mathbb{R}^{N \times D}$ for the GLUE tasks SST-2, CoLA, MNLI, QQP, RTE, and MRPC. We then fit $\mathbf{H}^{(i)}$ to $\mathbf{H}^{(j)}$ with an affine map and report the goodness of fit through the max error L2 norm, i.e., an approximation of $d(\mathbf{H}^{(j)}, \mathbf{H}^{(i)})$ on row $i$ and column $j$ of the grid.

