# OpenReview forum: "On Affine Homotopy between Language Encoders"
_NeurIPS.cc/2024/Conference — NeurIPS 2024 poster_

### Official Review · Reviewer_sF8t · 2024-07-06

**Soundness:** 2
**Presentation:** 2
**Contribution:** 1
**Rating:** 3
**Confidence:** 3

**Summary:**

The paper introduces formal mathematical concepts of intrinsic and extrinsic homotopy between language encoders. The first concept compares the behavior of two encoders on a concrete dataset, while the second concept compares them independently from the concrete dataset. The paper also demonstrates how to apply these concepts to measure the difference between different versions of the pre-trained BERT model.

In more detail, chapter by chapter:

- The 1st chapter introduces the problem of comparison of text encoders.
- The 2nd chapter is devoted to the discussion of the formal definition of a language encoder. The authors define a language encoder $\mathbf{h}$ very generally: as a function from all possible strings of some alphabet $\Sigma$ to a $d$-dimensional vector space $V$. They mention some ways to train an encoder in practice; however, for the rest of the theoretical part of the paper, $\mathbf{h}$ remains general enough and is not required to be implemented through a neural network.
- The 3rd chapter discusses the formalism of hemi-metrics and introduces uniform convergence norms on the space of language encoders. The uniform convergence norm allows the authors to introduce an affine alignment measure that shows how aligned two encoders are with each other.
- The 4th and 5th chapters introduce the notion of Intrinsic and Extrinsic Affine Homotopy on encoders. It is based on notions of affine alignment measure from the previous chapter, and encoder rank.
- The 6th chapter is devoted to adapting the notions above to the real-life scenario, where we can't check the encoder performance on all possible strings of our alphabet. Finally, asymmetric extrinsic and intrinsic similarity measures, that are usable in practice, are derived.
- In the 7th chapter the authors measure an extrinsic and intrinsic similarity between pre-trained BERT models, all trained with similar hyper-parameters but differing in random weight initialization and shuffling of training data (see "The MultiBERTs: BERT Reproductions for Robustness Analysis" paper). They show that these pre-trained BERT models can be differentiated from each other by the methods, introduced in the paper. They also provide a table of the correlations between their notions of similarity and some previously introduced notions of similarity between encoders. All experiments were performed using SST-2 and MRPC datasets.
- Finally, in the 8th chapter they discuss relations between extrinsic and intrinsic dissimilarities, their asymmetry, and the finding that some BERTs are more similar to other BERTs by affine similarity in one direction, but not necessarily in another.

**Strengths:**

- The paper introduces a novel way to compare the language encoders;
- An interesting and non-trivial mathematical tools are utilized.

**Weaknesses:**

First of all, it looks like this paper is not a good fit for this venue by its very nature:
- It is unclear what is the contribution of this work to the field of Neural Information Processing Systems. The paper introduced many novel concepts, but most of these concepts, starting from the definition of the language encoder itself, are very general, so their connection with neural networks looks far-fetched. The usefulness and purpose of the experiments are also unclear (see "Questions" section).
- 9-page format is too small for this type of scientific work; authors had to overuse the Appendices to fit their work into limited space. When I was reading the paper, I had to constantly go around in circles, moving between Appendix D ("Addenda on Affine Homotopy") and the main part of the paper.

Second, there are some general problems with the research and text:
- The experimental section in general is very limited. Only various versions of the BERT models are compared with each other, and only two datasets are considered. In line 235 there is a note that experiments show some task-independency, but how can we say anything about task-independency when we have only two tasks on hands?
- The conclusion and findings of the paper are also unclear (see "Questions" section).
- The paper is generally hard to read and understand. As was told above, this problem is partially connected with the unsuitable format; however, it is also connected with typos (see below) and lack of needed definitions (see "Questions" section).

**Questions:**

Questions:

- Why is your mathematical tool of checking, whether two encoders are similar to each other (line 153), called "Affine homotopy"? What is the connection to the notion of homotopy that we know from algebraic geometry?
- You found that some BERTs are more similar to other BERTs by affine similarity in one direction, but not in another, and that this fact says something about the "universality" of BERTs, that are easier to map "from". Can you please elaborate on this conclusion about their universality?
- By which machine learning algorithm is $\mathbf{Ð}$ approximated and how?
- What was the point of defining a dataset-independent way to measure the difference between encoders, if, starting from section 6, everything, in essence, became dataset-dependent again? (and of course, all experiments were dataset-depended as well)
- What is the purpose of the Figure 2? What can it tell about MULTIBERT encoders number $1, ... , 25$?
- In general, your paper doesn't explain what is the exact difference between, let's say, MULTIBERT encoder number $1$ and number $2$ (except that they are both BERTs with different weights init, etc). So what's the point of caring about their comparison in the first place? What useful conclusion can we make from such a comparison?
- What is the purpose of Table 1? If I understand correctly, it shows a correlation between different similarity measures, but what conclusion should we make from it?
- The most torturing question: what are $E_x$ and $E_y$, introduced in line 103, but never explained? I had to assume that they were just some neighborhoods of $x$ and $y$ to somehow get through the rest of the paper, but I'm not sure it's true.

Suggestions:
- I would suggest experimentally measuring the similarity of BERT with encoders of different natures, e.g. AlBERT, ELECTRA, and maybe even some LSTM-based encoders. Maybe the encoders of similar architecture (or trained on similar data) will be more similar by your similarity measures, than the encoders of very different architecture (or trained on very different data)? Or maybe not?
- I would suggest using more datasets for any further experiments.
- You heavily cite the "Metrics, quasi-metrics, hemi-metrics" chapter from the book called "Non-Hausdorff Topology and Domain Theory". However, this book is not well-known outside of the professional mathematical community. So, I would suggest explaining the conceptions from the book in more detail and providing some examples of hemi-metrics that would show why one wants to use hemi-metric instead of symmetric metric.
- Finally, I would suggest turning this work into a **journal paper** (after a revision) and merging Appendices D and B ("Additional Related Work") with the main part of the paper to make it more readable.

---

Typos and presentation issues:

- In 1st page, footnote №4, typo:
> In principle, one could relax the replace R^d with any finite-dimensional vector space
- In line 63, you wrote:
> There are two common ways that language encoders are created. The first is through autoregressive language modeling.

And the second one? Did you mean Masked Language Modelling?
- In line 119: typo:
> Let GL(V) We write
- In line 119, you introduced GL(V) without definition. I've supposed it's a General Linear Group for the rest of the paper.

**Limitations:**

The authors addressed the limitations of the theoretical part of their paper. I'd suggest adding the limitations of the experimental section (one architecture, a few datasets) as well (see above).

---

> ### Author Rebuttal · Authors · 2024-08-07
>
> Thank you for taking the time to read and review our work – we are grateful for all the suggestions. Further, we are happy to hear that our usage of novel non-trivial mathematical tools for the analysis of the space of encoders is appreciated. Below, we address the concerns raised in the review:
>
> - **“It is unclear what is the contribution of this work to the field of NeurIPS”**: Although we recognize that our usage of non-standard mathematical tools for NeurIPS, this is an attempt to more rigorously characterize the space of encoders as what they are – functions between countably infinite sets – and derive formal bounds that have concrete practical implications on downstream transfer learning behavior – and to that end, such tools are required. We further note that this paper follows a line of work, published at NeurIPS, on the similarity of neural representations (e.g., Boix-Adsera et al. (2022), Ding et al. (2021)) with extensions to the formalization of the problem as well as the objective of providing guarantees for extrinsic, downstream task behavior.
>
> ---
>
> - **format, “hard to read and understand”**: We retain only the relevant interim results in the main text that contribute to the main results of the paper, leaving the proofs to interested readers in the appendix. We acknowledge the resulting density of §2-5 and would use any additional space to add examples and proof outlines to support better understandability for the broader ML community.
>
> ---
>
> - **“Only various versions of the BERT models are compared with each other”, “What's the point of caring about [MultiBERTs’] comparison in the first place?”**.
> As found in the original paper and as seen in the experiments in §6, MultiBERTs may behave very differently among themselves. More concretely, MultiBERTs show vastly different downstream task performance (Sellam et al. 2022) and produce representation matrices of different ranks to precision $\epsilon$ (cf. §6). These are important attributes of encoders that we aim to capture in our notion of similarity, which are often disregarded (as discussed in §8). As such, MultiBERTs offer themselves well to investigate which task-independent (intrinsic) relation between the encoders best captures extrinsic dissimilarity in practice – the subject of our study in §7. Still, we acknowledge that a study across different architectures would yield broader comparisons, and we plan on including such a study as well as more extensive dataset coverage to complement our current experimental section.
>
> ---
>
> - **Connection to homotopy in algebraic geometry**: In classical algebraic topology, homotopy describes the continuous deformation of one function into another, typically parameterized by the interval $[0, 1]$. However, in this paper, we consider the concept of homotopy by using a broader set of parameterizations, denoted by $S$. This allows for more flexible transformations suited to specific classification problems. For instance, $S$ could be an affine space $Aff(V), Aff(V,W)$ or others, enabling more adaptable frameworks for continuous transformations.
>
> ---
>
> - **“Universality of BERT”**: From Theorem 4.1, we derive the influence of the encoder rank on affine mappability. We confirm that affine mappability is strongly influenced by the encoder rank in our experiments in Appendix G. This leads to the conclusion discussed in §7, where we find that some encoders seem to learn lower rank representation matrices (to precision $\epsilon$), and this correlate significantly with how easily one can affinely map to that encoder, which influences their concrete utility for transfer learning by Lemma 5.1.1 and Def. 5.2. Considerations about the asymmetry are discussed separately in §8.
>
> ---
>
> - **“How is Ð approximated”**: Ð is implemented as a gradient descent over affine maps (as mentioned in §6), where the loss is simply the max loss over all strings in the dataset. The gradient over the max is handled through subgradients in standard pytorch.
>
> ---
>
> - **“Task-dependence of experiments”**: This is a valid concern and a limitation of the work, which we partially acknowledge in the “Limitations” section. We note, however, that the formalism of a language encoder is by definition a map between countably infinite sets (strings to vectors), and our work makes a first attempt to derive properties of such a construction, which, evidently, also have practical implications for the encoders' use in transfer learning (cf. Discussion and §7). The approximation to the finite-string setting both shows the applicability of theoretical results on real datasets and allows us to draw comparisons to existing methods in representational similarity.
>
> ---
>
> - **“Purpose of Figure 2”**: As described in §3, Figure 2 plots intrinsic and extrinsic similarity, and, on a larger scale, shows an empirical approximation to the theoretical linear bound derived in Lemma 5.1. This does not tell us much about the encoders individually, but may act as a visual aid in seeing the empirical implications of the connections between our notions of intrinsic and extrinsic similarity.
>
> ---
>
> - **“Purpose of Table 1”**: Correct. It shows the correlation between affine intrinsic similarity measures (cols) and extrinsic measures of similarity (rows). As in the experimental setup in Ding et al. (2021), this measures how strongly differences in extrinsic similarity are picked up by the intrinsic similarity measures. We find that Ð (as well as other linear alignment methods) tend to be strongly (and significantly) indicative of the extrinsic, downstream behavior, as theorized by the upper bound in Lemma 5.1.1.
>
> ---
>
> - **“What are Ex and Ey”**: equivalent to the set E defined in Def. 3.3, Ex and Ey are non-empty subsets of X, and Y, respectively. We will add a comment, thanks!
>
> We again thank the reviewer for their suggestions and hope that we could address most concerns and open questions about the soundness and overall contribution of the paper.

---

> > ### Comment · Area_Chair_Refc · 2024-08-13
> > **Please respond to authors' rebuttal**
> >
> > Dear Reviewer sF8t,
> >
> > Thanks for your review. The authors have replied to your comment. Please engage in the discussion. After reading their rebuttal and other reviewers' feedback. Are you keeping the score unchanged and would you like to change your score?
> >
> > Thanks
> > AC

---

### Official Review · Reviewer_nK2G · 2024-07-09

**Soundness:** 4
**Presentation:** 3
**Contribution:** 4
**Rating:** 7
**Confidence:** 4

**Summary:**

In this paper, the authors study a nature question "What does it mean for two encoders to be similar" and proposed an intrinsic measure of similarity that aligns with extrinsic performance on downstream tasks. It introduces the concept of affine alignment and explores its properties and implications for understanding encoder relationships.
The main contributions are as follows:
1. The authors first define an metric space on encoders.
2. The authors extend the definition to account for transformations in a broad framework of $S$-homotopy for a set of transformations $S$.
3. As a concrete application of the framework, the authors study affine homotopy—the similarity for affine transformations.

**Strengths:**

1. The idea of the paper is quite novelty. Homotopy is a very important tool in algebraic topology. The application of homotopy theory in machine learning is very attractive.
2. The authors first define an (extended) metric space on encoders and then extend this definition to account for transformations in a broad  framework of $S$-homotopy for a set of transformations $S$.

**Weaknesses:**

1. The motivation for introducing affine Hemi-Metrics to measure the similarity of two encoders is not so clear to me.
2. The definition of Extrinsic Homotopy seems not like a mathematical definition since authors does not claim that how to quantify the performance.
3. More experiments are needed for supporting the claim in the paper.
4. Such method seems not practical.

**Questions:**

1. Please explain clearly the motivation why we need Hemi-metrics, I think maybe from both sides of topological view and machine learning view.
2. Please explain the connection between intrinsic homotopy and extrinsic homotopy. I think it can help us understand theorem 5.1 better.

**Limitations:**

See the weaknesses.

---

> ### Author Rebuttal · Authors · 2024-08-07
>
> We thank the reviewer for their thorough review and helpful feedback. We are pleased to read that you appreciate the novelty of the approach and the breadth of the ideas introduced in the paper. Below, we address the specific concerns raised in the review:
>
>
> 1. **“Elaborate your Motivation of using Hemi-Metrics”**: Although some previous work has explored measuring similarity in proper metric spaces, we posit that purely from a machine learning perspective, there is an asymmetry to the problem of measuring the similarity between encoders. More concretely, when we assess the similarity of two encoders in terms of how closely we can affinely map their classification probabilities on a task, we find this to be asymmetric theoretically as well as practically. Some encoders may be more powerful than others (e.g., think of a lower-rank encoder, such as the ones generated in Appendix G). A symmetric distance, therefore, is not practical as it does not capture this phenomenon. This motivates using Hemi-Metrics.
>
> ---
>
> 2. **“Extrinsic Homotopy does not seem like a mathematical definition”**: We agree that Definition 5.1 is somewhat informal. We will explicitly state this in the next revision – thank you for pointing this out. We note, however, that we provide a formal mathematical definition of extrinsic homotopy in Lemma 5.2 similar to intrinsic affine homotopy in Def. 4.2.
>
> ---
>
> 3. **“More experiments are needed to support the claim”**: We would like to highlight that most theoretical results/claims (Thm. 4.1, Lemma 5.1, and Thm. 5.1) were evaluated empirically across a large number of encoders that exhibit significant differences in properties and downstream task behavior (cf. §6, [1], [2]). Still, in light of this and other reviews, we acknowledge that an evaluation across more datasets may help support the claim, which we will include for a next revision – we thank the reviewer for the suggestion!
>
> ---
>
> 4. **“Such method seems not practical”**: Although we acknowledge the gap between some properties derived in §2-5 with the experimental results (see “Limitations” Appendix A), our derivation of intrinsic similarity measures that upper bound extrinsic dissimilarity (Lemma 5.1, Def. 5.2) is purely practically driven. Namely, we show that our measure of intrinsic similarity is indicative of how an encoder may perform across downstream tasks in the transfer learning setting. Further, our discussion of the algebraic properties of affine homotopy gives us a rich theoretical foundation useful to understand phenomena we observe empirically (cf. §7-8), such as:
>
> - The theoretical upper linear bounds of intrinsic similarity on task performance (Lemma 5.1, Def. 5.2) surface empirically, and therefore show us how our intrinsic measures of similarity can indicate downstream task performance similarity across arbitrary tasks. This is of significant practical interest for encoders’ utility in transfer learning, as this may reduce the need to do task-specific evaluation, as motivated in §1.
>
> - Our proof about how affine mappability is affected by the encoder rank surfaces in our experiments §7 - “The Influence of Encoder Rank deficiency” as well as in our additional experiments in Appendix G.
>
> ---
> Questions
> ----
>
>
> 1. See point 1. above
>
> ---
>
> 2. **“Explain the connection between intrinsic homotopy and extrinsic homotopy [Thm. 5.1]”**: We prove in Lemma 5.1 that for some fixed linear classifier $\psi’$, the extrinsic dissimilarity (i.e., how closely we can map the output probabilities of representations of encoder h with the ones from encoder g) is linearly bounded by the intrinsic similarity measure (Eq. 9b). Theorem 5.1 makes a stronger statement and states that the Haussdorff-Hoare variant of the intrinsic distance upper bounds the extrinsic dissimilarity over all possible linear classifiers. This shows that our methods of measuring intrinsic homotopy linearly upper bounds, and is therefore indicative of extrinsic homotopy. The practical implications of this are significant, as this shows we can derive encoder similarity measures that are indicative of their downstream task behavior, without the need for task-specific evaluation.
>
> We again thank you for the insightful suggestions and hope that our clarifications address your concerns about the overall contributions of our work.
>
> ---
>
> [1] Jesse Dodge, Gabriel Ilharco, Roy Schwartz, Ali Farhadi, Hannaneh Hajishirzi, and Noah Smith. 2020. Fine-tuning pretrained language models: Weight initializations, data orders, and earlystopping.
>
> [2] Thibault Sellam, Steve Yadlowsky, Jason Wei, Naomi Saphra, Alexander D’Amour, Tal Linzen, Jasmijn Bastings, Iulia Turc, Jacob Eisenstein, Dipanjan Das, et al. "The MultiBERTs: BERT reproductions for robustness analysis." In ICLR, 2022.

---

> > ### Comment · Reviewer_nK2G · 2024-08-09
> >
> > Thank you for authors' reply which answer my part of questions. But it still makes me unclear the connection between intrinsic and extrinsic homotopy since you also mentioned the similarity between Lemma 5.1 and Def 4.2. I am wondering if I understand it right that extrinsic homotopy is the upper bound of intrinsic homotopy?
> >
> > Also, for the motivation of Hemi-Metric,  you also mentioneed "a symmetric distance, therefore, is not practical as it does not capture this phenomenon. " But the Hemi-Metric is symmetric, or a metric should be symmetric, right? I may not get your point, please explain it clearly?

---

> > > ### Author Response · Authors · 2024-08-12
> > > **Replying to Official Comment by Reviewer nK2G**
> > >
> > > Thank you for taking the time to read our response and reply.
> > > - **"I am wondering if I understand it right that extrinsic homotopy is the upper bound of intrinsic homotopy?"**: No, Lemma 5.1.1 and Theorem 5.1 both provide a linear upper bound on **extrinsic** affine homotopy measures by **intrinsic** ones (not the other way around). More specifically, the notions of intrinsic homotopy provide an upper bound of specific extrinsic ones for a fixed task; Lemma 5.1.1 and the worst-case task; Thm. 5.1. Note that in this way (and not the other) we can make guarantees about (extrinsic) downstream task performance dissimilarity from intrinsic measures of similarity — which is what we aim to achieve.
> > > ---
> > > - **"But the Hemi-Metric is symmetric, or a metric should be symmetric, right? I may not get your point, please explain it clearly?"** No, compared to metrics, hemi-metrics are, by definition, not symmetric (see Def. 3.2; "Hemi-Metrics", in contrast to Def. 3.1; "Extended Metrics"). Our point is that to derive the intrinsic similarity between encoders that is indicative (cf. Lemma 5.1.1, for instance) of their extrinsic similarity ( = the closeness with which another encoder’s task classfication probabilities can be matched affinely), we do not want symmetric measures, as this would not capture the directionality of the problem. The directionality in this context means that one encoder may be more powerful than another as you can affinely match another encoder’s output probabilities closely, whereas the inverse may not be possible (cf. experiments on rank-deficient encoders, Appendix G).
> > >
> > > We hope this helped clarify your concerns!

---

> > > > ### Comment · Reviewer_nK2G · 2024-08-12
> > > >
> > > > Thank you for your reply! Your explaination solves my concerns.

---

### Official Review · Reviewer_6x9V · 2024-07-12

**Soundness:** 3
**Presentation:** 4
**Contribution:** 3
**Rating:** 6
**Confidence:** 3

**Summary:**

The paper aims to formally define, derive and then analyse similarity between pretrained language encoders, focusing on aspects of intrinsic (task-independent) similarity and extrinsic (task performance-oriented) similarity. The paper is mostly of theoretical nature, aiming to properly and formally define the studied aspects of similarity and then aiming to propose the idea of transformations in an (affine) homotopic framework. The work makes a step towards more formal studies of representation similarity within language encoders (and probably decoder-style LLMs in future work).

The main non-theoretical finding, based on experiments with MultiBERT models (i.e., BERT models trained from different random seeds) is that there exists (as expected) a correlation between the defined intrinsic and extrinsic notions of similarity.

**Strengths:**

- The paper provides a fresh perspective on the question of similarity between language encoders (and language models) in general, aiming to rigorously formalise and derive different properties associated with intrinsic and extrinsic similarity.

- The paper is quite dense but admirably well-written given its largely formal and 'math-heavy' content. I see it also as a potentially very didactic piece of work which could inspire additional work in this space.

- Related work, limitations, implications of the key results (both from the theoretical as well as from the more practical perspectives) are all very comprehensive and nicely structured.

**Weaknesses:**

- The work is heavily focused on theory and theoretical contributions; this means that its more practical findings are the weaker part of the work and the experimental setup and results are a bit underwhelming:
a) The only studied architecture is the BERT architecture, and the work just aims to quantify correlation between intrinsic and extrinsic similarity for MultiBERT models (which just use different random seeds). As far as I am concerned, a positive correlation between intrinsic and extrinsic similarity is very much expected for this group of models.
b) The work should ideally study other encoder architectures and aim to establish how the notion of similarity changes over the spectrum of 'expected model distance' and what implications it might have. Speaking of the 'expected model distance', what one could/should study here is:
-- models of the same architecture starting from the same seed where different checkpoints were taken
-- models of the same architecture with different random seeds (this is the only thing studied in the paper atm)
-- models of similar architecture but not completely the same (e.g., BERT vs RoBERTa)
-- completely different architectures (e.g., BERT vs tELECTRA)
c) The work also focuses on very basic GLUE tasks which also limits the generalisability of the main findings, and additional experiments over tasks of different complexity are required here to fully trust the core ideas (which would also increase the impact of the work substantially imo).

- While mathematical rigor in the paper is very useful, the paper would contribute from a short (sub)section aiming to properly 'distill' the key take-home messages in a plain(er) language so that it also becomes more obvious how future work could contribute from the more theoretical insights (e.g., can the derived measures be used for computing representation similarity in general)? That would be a very useful addition to the work.

**Questions:**

- Beyond page limit, is there any other reason why only two tasks (and very simple ones) are targeted for the main experiments?

- Have the authors considered running the similarity analyses also with BERT models having different hyper-parameters? Can we expect the affine homotopy properties to hold for such models? What implications might this finding have?

- I am missing the reason why certain encoders end up being more informative than others. How do you define 'being informative' in this context? Why is it important?

- Would it be possible to derive similar measures for decoder-only LMs in the future?

**Limitations:**

Limitations are (to most part) properly discussed in Appendix A. Some additional reflection on the current limitations of experiments (and practical aspects of the work) might be necessary and useful.

---

> ### Author Rebuttal · Authors · 2024-08-07
>
> We thank the reviewer for taking the time to read our work, as well as for their thorough review and helpful feedback. We are very happy to hear the positive comments about our formalization of the problem and our writing. We address the open points and concerns raised in the review below:
>
> - **"The work should ideally study other encoder architectures and aim to establish how the notion of similarity changes over the spectrum of 'expected model distance' and what implications it might have"** We fully agree that such study across encoder architectures and additional tasks would complement our current experiments and we plan to add this for the next revision of the paper – thank you for the suggestion!
>
> ---
>
> - **"The only studied architecture is the BERT architecture [...] which just use different random seeds"**: Although we fully agree with the point raised in b) to get a broader coverage of results by further studying different encoder architectures and running experiments across more datasets (which was solely limited by page limits), we would like to restate the motivation and significance of the found results mentioned in §1. Namely, as found in the original paper and as seen in the experiments in §6, MultiBERTs may behave very differently among themselves. More concretely, MultiBERTs have been shown to produce embeddings that yield significantly different downstream task performance (Dodge et al. 2021 [1], Sellam et al. 2022, [2]) and produce representation matrices of different ranks to precision $\epsilon$ (cf. §6). These are important attributes of encoders that we aim to capture in our notion of similarity, which are often disregarded (as discussed in §8). As such, MultiBERTs offer themselves well to investigate which task-independent (intrinsic) relation between the encoders best captures extrinsic dissimilarity in practice – the subject of our study in §7.
>
> ---
>
> - **“A positive correlation between intrinsic and extrinsic similarity is very much expected for this group of models”**: our derivation exactly shows that specifically our notions of intrinsic similarity will be correlated with the extrinsic similarity **independent** of the considered group of models. The linear upper bound on the extrinsic dissimilarity by our notion of intrinsic similarity does not make any assumptions about the nature of the encoder, and we therefore expect such correlations to hold across encoder families.
>
> ---
>
> - **“Have the authors considered running the similarity analyses also with BERT models having different hyper-parameters?”** As mentioned above, we did not make assumptions about the underlying encoder structure to derive the upper bound on the extrinsic similarity. Although we expect gaps in similarity to become more prominent, we do not expect these to affect the correlation between intrinsic and extrinsic similarity.
>
> ---
>
> - **“I am missing the reason why certain encoders end up being more informative than others.”** We refer to the fact that in theory (cf. Thm. 4.1) higher-rank encoders are more powerful, as we can exactly affinely map from their image into the image of a lower-rank encoder, whereas the inverse does not hold. We also find this to surface empirically (cf. §7, “The Influence of Encoder Rank Deficiency”), and it has a significant impact on the affine mappability between the representation spaces (intrinsic similarity), and, as a result, task performance and the extrinsic similarity.
>
> ---
>
> - **“Would it be possible to derive similar measures for decoder-only LMs in the future”** Although our method discusses encoder functions motivated by their usage for transfer learning, we do not in principle make assumptions about the nature of the function. In that sense, decoder representations may be evaluated equivalently.
>
> We again thank you for the insightful suggestions and hope that our clarifications address any concerns about the overall contributions of our work.
>
> ---
>
> [1] Jesse Dodge, Gabriel Ilharco, Roy Schwartz, Ali Farhadi, Hannaneh Hajishirzi, and Noah Smith. 2020. Fine-tuning pretrained language models: Weight initializations, data orders, and earlystopping.
>
> [2] Thibault Sellam, Steve Yadlowsky, Jason Wei, Naomi Saphra, Alexander D’Amour, Tal Linzen, Jasmijn Bastings, Iulia Turc, Jacob Eisenstein, Dipanjan Das, et al. "The MultiBERTs: BERT reproductions for robustness analysis." In ICLR, 2022.

---

> > ### Comment · Reviewer_6x9V · 2024-08-13
> > **I appreciate the response...**
> >
> > ...and the clarifications provided. I would keep the current score, as my questions were mostly resolved 'theoretically' without providing additional empirical evidence on some of the claims from the responses.

---

### Official Review · Reviewer_Jeko · 2024-07-14

**Soundness:** 3
**Presentation:** 2
**Contribution:** 2
**Rating:** 6
**Confidence:** 1

**Summary:**

This paper presents theoretical analysis on "intrinsic alignment" of two or more pretrained language encoders (e.g. BERT trained with various seeds). The work proposes computing the intrinsic alignment between two encoders by first defining a algebraic metric space in which these two encoders exist, and then looking at the affine homotopic transformations that are possible in the given space. The primary motivation of the work is to set theoretical bounds on the similarity of two given encoders, and for the intrinsic alignment to have a strong (positive) correlation to extrinsic alignment (in simple terms, whether two encoders produce similar outputs for similar inputs). The work claims that this is important to have an "elementary understanding" of these encoders, and that having these theoretical guarantees can help derive a richer set of properties of the relationship between encoders.

The work first proposes the methodology to compute these alignments, and then conducts experiments on 25 multiBERT encoders, which are BERT models trained with varying seeds. Two GLUE classification tasks are used, SST-2 and MRPC. The results show that that such an alignment can indeed be computed, and that there is a positive correlation between the intrinsic and extrinsic alignments.

**Strengths:**

- The work shown in the paper approaches some of the nuances of modern LM training (randomization, seeds etc) from a theoretical perspective, which can not only help solidify our understanding but also give us concrete bounds and limitations when these models are trained
- The results and discussion emerging from the analysis seems sound, and something that can potentially advance our understanding of these encoders further

**Weaknesses:**

- While the paper makes some effort to depict the practical value of the underlying method, it falls short of giving actual examples. For instance, the paper mentions that deriving intrinsic alignment can help discover "richer" properties - but for a reader (that does not necessarily have theoretically expertise), it is difficult to see what these richer properties look like
- There is also very little discussion of practical aspects of running this analysis; The appendix mentions that intrinsic alignment is more expensive to compute than extrinsic alignment; given that the aim of the work is to set upper bounds, why not just compute the extrinsic scores? I was also unable to find any discussion on whether different kinds of encoders can be compared (which is easy to do for downstream tasks).

**Questions:**

- What are some of these "richer" properties that intrinsic alignment can help discover?
- Can these be run across encoders with different architectures?
- The paper mentions that one advantage of intrinsic alignment is the possibility to define an order over the encoders. What does this imply?
- I am wondering if these alignment measures can somehow be used to improve pre-training; lets say if we can compute the similarity between a well trained model and one that is in the training loop, and make training decisions based on how close/far the alignment is. Do you envision any such use of the proposed metric?
- Is there a way to convert the alignment metric into a measure of "badness" for a given encoder? I envision it would be useful to weed out "bad" models than to just compare "good" ones.

**Limitations:**

Authors have addressed limitations adequately.

---

> ### Author Rebuttal · Authors · 2024-08-07
>
> We thank the reviewer for their thorough review and the helpful feedback and ideas they provided! We are happy to hear that our theoretical results as well as practical implications are appreciated. In the following, we address the open points and concerns:
>
> - **“It is difficult to see what the richer properties are, but for a reader (that does not necessarily have theoretically expertise), it is difficult to see what these richer properties look like”**: What is meant by “richer properties” in the introduction refers to our much more general theoretical results and the empirical implications that result from formalizing and studying encoders as what they are – functions between countably infinite sets. For instance, our derivation for Remark 3.2 finds that only one-sided affine alignment (Eq. 9.2) is non-trivial, and further, defining hemi-metric spaces allows us to derive proper mathematical preorders and equivalence relations on the space of encoders, which (e.g., Thm. 4.1) has concrete empirical implications (cf. §7, “The Influence of encoder Rank Deficiency”). Further, our formalization allows us to explicitly construct the linear bounds of extrinsic dissimilarity not just for a fixed classifier (Lemma 5.1.1), but for the “worst-case dissimilarity” (cf. Thm. 5.1) -- this highly practical, as it shows our intrinsic similarity measure to be indicative of how an encoder may perform across downstream tasks in the transfer learning setting. Still, we acknowledge the density of the theoretical sections, and as per recommendation by this reviewer and reviewer 6x9V we plan to create a distilled overview paragraph of the theoretical contributions in a next revision to appeal to the broader ML community.
> ---
> - **“There is also very little discussion of practical aspects of running this analysis, The appendix mentions that intrinsic alignment is more expensive to compute than extrinsic alignment; given that the aim of the work is to set upper bounds, why not just compute the extrinsic scores?”**: The practicality comes exactly *from* the fact that we show that the intrinsic similarity measures can be already indicative of task performance – not just for a fixed task (Lemma 5.1.1.), but for the worst case task across all tasks (see Thm 5.1., as well as §8; “Implications of §5”). In other words, we show that computing our intrinsic measures of similarity may already indicate the overall utility of an encoder for *any* downstream task, thus potentially eliminating the need to do task-specific evaluation of extrinsic scores.
> ---
> - **“Lacking a discussion of what kinds of encoders can be compared”, “Can these be run across encoders with different architectures?”**: Generally, we do not make any assumptions about the structure of the encoder in the derivation of our theoretical results, resulting in wide generalizability of our method. In other words, our method can be applied to measure the (intrinsic) similarity between *any* two encoders and evaluate how this affects how (extrinsically) similar they can be in terms of their output probabilities.
> ---
> - **“What is the benefit of defining an order over encoders”, “alignment metric into a measure of "badness" for a given encoder”**: Purely mathematically speaking, a proper order over a set is a powerful structure that allows us to define proper equivalence relations over encoders. Beyond equivalence, the order between encoders (cf. Lemma 4.1.) indicates that we may be able to exactly affinely map from one image into the image of a lower-rank encoder, whereas the inverse does not hold. A higher-rank encoder is therefore more powerful, and intrinsic affine homotopy is indicative of this. We find this rank deficiency to surface empirically (cf. §7, “The Influence of Encoder Rank Deficiency”). By Lemma 5.1.1., this also strongly affects extrinsic similarity, i.e., how closely we can affinely reach the output probabilities of an encoder for a specific task from another encoder – yielding a measure of the “goodness”/”badness” of an encoder, in your words.
> ---
> - **“these alignment measures can somehow be used to improve pre-training”**: this is an interesting suggestion, especially in light of the nuances in fine-tuning that have shown to have large effects on downstream task performance of the pretrained encoders [1]. We can add this as possible future work or motivation, thank you for the suggestion.
>
> We again thank the reviewer for the insightful suggestions and hope that our clarifications address their concerns about the overall contributions of our work!
>
> ---
>
> [1] Jesse Dodge, Gabriel Ilharco, Roy Schwartz, Ali Farhadi, Hannaneh Hajishirzi, and Noah Smith. 2020. Fine-tuning pretrained language models: Weight initializations, data orders, and earlystopping.

---

> > ### Comment · Reviewer_Jeko · 2024-08-13
> >
> > I thank the authors for the response, it has helped me understand some of the concepts better. I still feel like the practical aspects are still unclear to me (e.g. the tradeoff of runtime vs guarantees, the implications of "good"/"bad" encoders), however, this may be because I am not very well versed on the theoretical side of algebraic spaces. I have gone through the other reviews, and will maintain my score.

---

### Comment · Area_Chair_Refc · 2024-08-08
**Please respond to authors' rebuttals**

Dear Reviewers,

Thanks for writing your reviewers of the paper. Now the authors' rebuttals are in. Please go through them and see if they have addressed your questions. Please start discussions with the authors if you have further comments.

Regards,
AC

---

### Decision · Program_Chairs · 2024-09-25

**Decision:**

Accept (poster)

**Comment:**

This paper presents theoretical analysis on "intrinsic alignment" of two or more pretrained language encoders. As pointed out by the reviewers the theoretical analysis in the paper is very solid and well presented. This kind of work is very valuable for the community to understand the neural encoders and improve them further. A weakness that all the reviewers mentioned is a lack of thorough experiments. For the camera ready version of the paper, I highly encourage the authors to add more experiments to study other encoders as promised in the rebuttal. This will make the paper stronger.